# Chemical perturbation of an intrinsically disordered region of TFIID distinguishes two modes of transcription initiation

**Zhengjian Zhang[1]\*, Zarko Boskovic[2,3], Mahmud M Hussain[2,3], Wenxin Hu[1], Carla Inouye[4], Han-Je Kim[3], A Katherine Abole[5], Mary K Doud[3], Timothy A Lewis[3], Angela N Koehler[3,6], Stuart L Schreiber[2,3], Robert Tjian[1,4]**

[1]Transcription Imaging Consortium, Janelia Research Campus, Howard Hughes Medical Institute, Ashburn, United States; [2]Department of Chemistry and Chemical Biology, Howard Hughes Medical Institute, Harvard University, Cambridge, United States; [3]Center for the Science of Therapeutics, Broad Institute, Cambridge, United States; [4]Li Ka Shing Center for Biomedical and Health Sciences, Department of Molecular and Cell Biology, Howard Hughes Medical Institute, University of California, Berkeley, Berkeley, United States; [5]Department of Chemistry, University of California, Berkeley, Berkeley, United States; [6]David H. Koch Institute for Integrative Cancer Research, Department of Biological Engineering, Massachusetts Institute of Technology, Cambridge, United States

**Abstract** Intrinsically disordered proteins/regions (IDPs/IDRs) are proteins or peptide segments that fail to form stable 3-dimensional structures in the absence of partner proteins. They are abundant in eukaryotic proteomes and are often associated with human diseases, but their biological functions have been elusive to study. In this study, we report the identification of a tin(IV) oxochloride-derived cluster that binds an evolutionarily conserved IDR within the metazoan TFIID transcription complex. Binding arrests an isomerization of promoter-bound TFIID that is required for the engagement of Pol II during the first (de novo) round of transcription initiation. However, the specific chemical probe does not affect reinitiation, which requires the re-entry of Pol II, thus, mechanistically distinguishing these two modes of transcription initiation. This work also suggests a new avenue for targeting the elusive IDRs by harnessing certain features of metal-based complexes for mechanistic studies, and for the development of novel pharmaceutical interventions.

\*For correspondence: zhangzh@ janelia.hhmi.org

## Introduction

Intrinsically disordered proteins/regions (IDPs/IDRs) constitute a significant fraction of the metazoan proteome (*Liu et al., 2006*; *Uversky, 2013*). By virtue of their structural malleability and propensity to interact with multiple-binding partners, these peptide stretches of ~30 or more amino acid residues have become increasingly recognized for their pivotal and prevalent role in cellular functions including many implicated in human disease pathogenesis (*Babu et al., 2011*). IDRs usually function through transient and weak interactions, rendering them difficult subjects for mechanistic studies.

IDRs are typically composed of low-complexity sequences and are often rich in polar amino acid residues, making them challenging targets for intervention by conventional small-molecule inhibitors, which often require stable hydrophobic-binding pockets (*Metallo, 2010*). However, conceptually, this feature of IDRs may be suitable for interactions with the hydrophilic and periodic metal–oxygen

**eLife digest** DNA contains instructions to make all the proteins and other molecules that drive essential processes in cells. To issue such specific sets of instructions, a section of DNA—called a gene—is first copied to make molecules of messenger ribonucleic acid (or mRNA for short) in a process called transcription. This process is tightly regulated in all living organisms so that only a subset of genes are actively transcribed at any time in a given cell.

A group or 'complex' of proteins called TFIID plays an essential role in starting the transcription of genes that encode proteins in humans and other eukaryotic organisms. However, it is tricky to study how TFIID works because mutant cells that are missing individual components of the complex are unable to properly transcribe the required genes and soon die. Consequently, many studies of TFIID have used purified proteins in artificial systems where it is possible to examine particular aspects of TFIID activity in depth.

Here, Zhang et al. used a combination of chemistry, biochemistry, and molecular biology techniques to identify a new molecule that can selectively bind to the TFIID complex. In an artificial system containing purified proteins and other molecules, this molecule 'locks' TFIID onto DNA and prevents a change in shape that is required for transcription to start. The experiments show that this rearrangement is only required to make the first mRNA copy of a gene because the molecule had no effect on initiating further rounds of transcription on the same DNA.

Zhang et al.'s findings reveal that TFIID is very dynamic in controlling transcription, and that subsequent rounds of transcription follow a different path to make mRNAs. The next steps are to use new techniques such as single-molecule imaging to directly visualize the molecules involved in transcription, and to use the new molecule to block the start of transcription in living cells.

backbones found in a group of metal clusters known as polyoxometalates (POMs). POMs, primarily composed of transition metals and oxygen, have been previously reported to exhibit potent biological activities (*Rhule et al., 1998*), although these activities are rather promiscuous due to POMs' limited structural diversity, large size, and dependence on charge-based interactions in protein binding (*Judd et al., 2001*; *Li et al., 2009*; *Geng et al., 2011*; *Narasimhan et al., 2011*). Efforts have been made to improve the specificity and potency of POMs (*Flutsch et al., 2011*; *Gao et al., 2014*), and it seems that biological application of this metal complex-based protein-targeting strategy could benefit from new chemistry beyond conventional POMs to gain greater chemical space.

Transcription initiation by eukaryotic RNA polymerase II (Pol II) is a highly regulated process requiring the coordinated actions of Pol II and a group of general transcription factors (GTFs, i.e., TFIIA, TFIIB, TFIID, TFIIE, TFIIF, and TFIIH) (*Roeder, 1996*). Of all the GTFs, TFIID is the one with intrinsic DNA sequence specificity and is responsible for nucleating the assembly of a preinitiation complex (PIC) at core promoters (*Roeder, 1996*). TFIID-promoter binding, which can be further stabilized by TFIIA, creates a platform for TFIIB loading, which in turn allows the engagement of TFIIF and Pol II. TFIIE and TFIIH are the last to join the PIC, after which promoter melting and RNA synthesis can occur. After Pol II leaves the promoter during the initial (de novo) round of transcription, a reinitiation scaffold containing TFIID has been proposed to remain on the DNA template and facilitate subsequent rounds of transcription (reinitiation) in reconstituted mammalian systems (*Hawley and Roeder, 1987*; *Zawel et al., 1995*).

TFIID is a protein complex composed of the TATA-binding protein (TBP) and ~14 TBP-associated factors (TAFs) (*Albright and Tjian, 2000*; *Matangkasombut et al., 2004*). TBP recognizes and binds to the TATA box, while TAF1 and TAF2 recognize the Initiator element (Inr), and the TAF6-TAF9 module recognizes the downstream core promoter element (DPE) (*Juven-Gershon and Kadonaga, 2010*). It is generally accepted that higher TFIID-promoter affinity leads to more robust transcription, and indeed it is thought that a primary role of sequence-specific activators is to recruit TFIID to promoters (*Roeder, 1996*; *Albright and Tjian, 2000*; *Matangkasombut et al., 2004*; *Juven-Gershon et al., 2006*). However, transcription initiation is a dynamic process and the release of at least a portion of the TFIID-promoter DNA contacts has been shown to be a critical step for productive initiation (*Yakovchuk et al., 2010*). In addition, TFIID may assume diverse structures and its recognition of core promoter elements can be modulated by activators and post translational modifications (*Lewis et al., 2005*;

*Juven-Gershon et al., 2008*; *Liu et al., 2009*; *Cianfrocco et al., 2013*). These distinct functional states of TFIID and their transitions are likely critical for gene-specific transcription regulation, but they are difficult to probe by conventional biochemistry and genetic analysis.

In a search for small molecule compounds to selectively perturb TFIID function, we identified a tin-based metal cluster as a specific binder and modulator targeting an IDR within the TAF2 subunit of metazoan TFIID. By virtue of its specificity for interfering with the first round of transcription initiation, this metal cluster compound serves as a useful tool for studying the role of TFIID in both transcription initiation and reinitiation. This non-POM metal cluster revealed a novel mode of interaction with a low-complexity protein domain, demonstrating the feasibility of using metal-based compounds to selectively target IDRs.

## Results

### Identification of a tin(IV) oxochloride-derived cluster as a TFIID inhibitor

We screened a library of ~10,000 organic compounds for binders to metazoan TFIID using a small-molecule microarray platform (*Casalena et al., 2012*). In this screen, chemicals of diverse structures were printed on a functionalized glass surface and the binding of TFIID was detected by specific antibodies (*Figure 1A*). We identified one compound (**1**, ChemDiv 7241-4207) that reproducibly and selectively bound to both *Drosophila* and human TFIID (*Figure 1B,C*). As controls, no binding to the antibodies or two other multi-subunit complexes of the human Pol II core transcription machinery, TFIIH and Pol II, was observed in counter screenings (*Figure 1B*).

To assess the effect of compound **1** on transcription, we developed an integrated functional assay consisting of a reconstituted human cell-free transcription system (*Figure 2A*). In this assay, a complete set of highly purified GTFs (TFIIB, TFIID, TFIIE, TFIIF, and TFIIH; TFIIA is not required) plus Pol II was incubated with the lead compound first, followed by incubation with a promoter-containing DNA template for transcription. As a control, TBP was used in place of TFIID to support a 'basal' transcription that also requires the rest of the protein factors. We found that the commercially supplied compound **1** (ChemDiv 7241-4207) inhibited both *Drosophila* and human TFIID-directed transcription, but not transcription directed by TBP (*Figure 2B* and *Figure 2—figure supplement 1A,B*), suggesting a TAF-specific mechanism of inhibition. Further characterization indicated that this inhibition (i) is sensitive to the dose of TFIID used in the reaction (*Figure 2—figure supplement 1B*), (ii) can be alleviated by the addition of more TFIID after chemical treatment, but not by the addition of any other protein factors (*Figure 2C*), confirming that TFIID is the most likely target of inhibition in the reaction, and (iii) is insensitive to various mutations in core promoter elements (*Figure 2—figure supplement 1C*). Taken together, our transcription results suggest that the inhibitory activity specifically targets an evolutionarily conserved TAF subunit of TFIID that is required for a basic function of TFIID during Pol II transcription initiation in vitro.

In an effort to perform a structure-activity relationship analysis, we resynthesized compound **1** in-house and were surprised to find that the resynthesized compound was completely inactive in the transcription assay (*Figure 3A*). By comparing three batches of an analog compound (**2**) with varying levels of inhibitory activity (*Figure 3—figure supplement 1*), we found that the inhibitory activity correlated with levels of a tin-containing material detected by elemental analysis (*Figure 3B*). This material is likely derived from tin(II) chloride (SnCl$_2$) added as an anti-oxidant in the final recrystallization step in a subset of the commercially supplied samples (*Figure 3—figure supplement 2*). After excluding most common tin-containing compounds as candidates, we found that tin(IV) oxochloride, prepared by any of several established routes (*Dehnicke, 1961*; *Messin and Janierdubry, 1979*; *Sakurada et al., 2000*), consistently reproduced the specific inhibition of TFIID-directed transcription (*Figure 3C* and *Figure 3—figure supplement 3*). Dose-response titration revealed a Hill coefficient of ~1, suggesting a non-cooperative binding of this chemical to its biological target (*Figure 3D*). This compound, which consists of tin, bridging oxygen, and chlorine ligands, may form ladder-like clustered structures and coordinate to atoms with lone electron pairs, such as the nitrogen in pyridine (*Dehnicke, 1961*; *Messin and Janierdubry, 1979*; *Holmes et al., 1987*; *Sakurada et al., 2000*), or as is perhaps more functionally relevant, the imidazole groups of histidine residues in proteins (*Figure 3—figure supplement 3E*). We concluded that the tin(IV) oxochloride-derived cluster is the ingredient within the active commercial supplies responsible for the TFIID-specific transcriptional inhibitory activity.

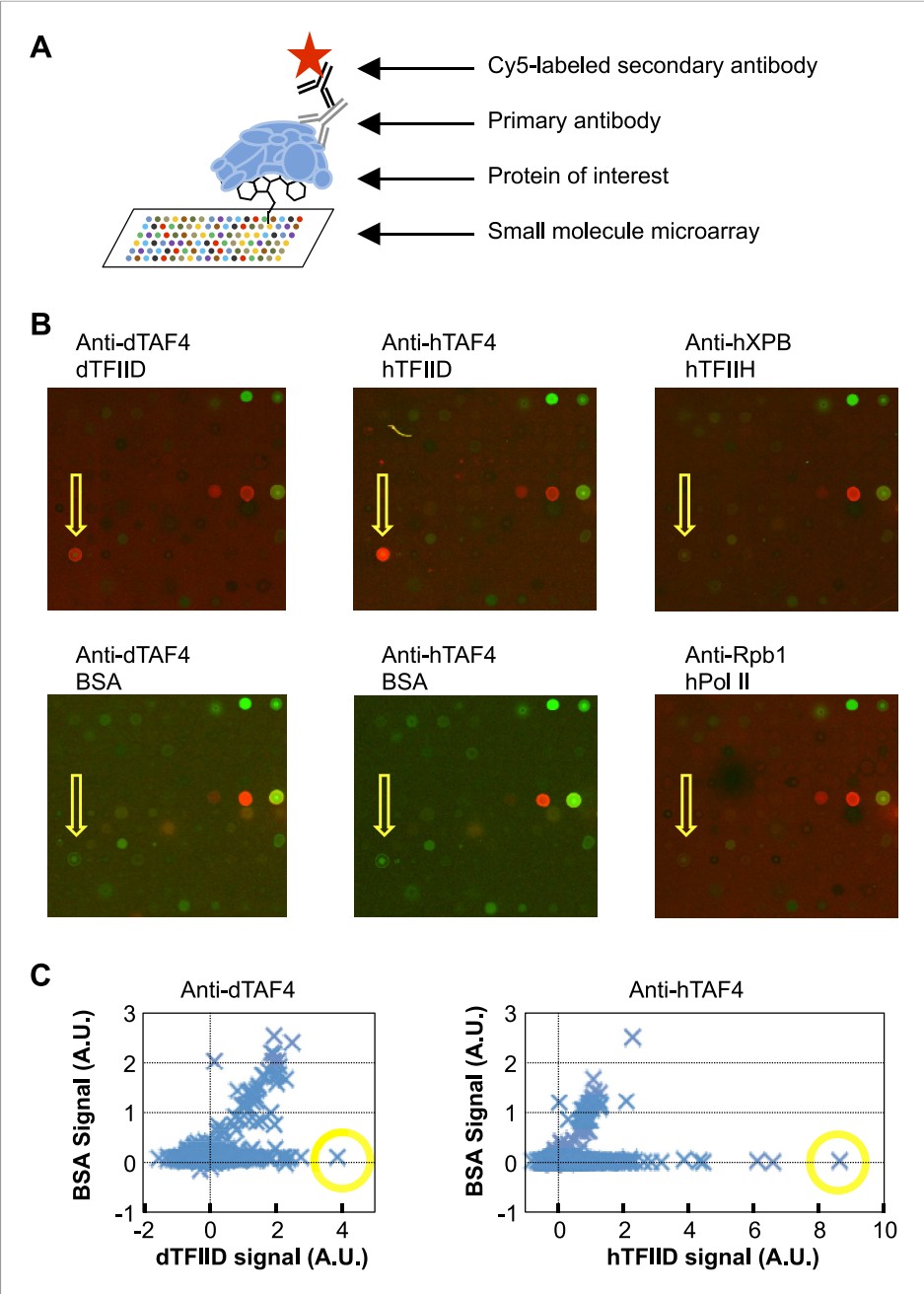

**Figure 1**. Small-molecule microarray screening for TFIID-specific modulators. (**A**) Screening schematic: chemicals of diverse structure were covalently attached to a functionalized glass surface, and incubated with protein of interest, the binding of which was indicated by primary antibodies recognized by a specific fluorescently (Cy5, red) labeled secondary body. (**B**) Representative images of an area of arrays probed with bovine serum albumin (BSA, control), Drosophila (d) TFIID, human (h) TFIID, TFIIH (control), and Pol II (control) in combination with specified primary antibodies. Yellow arrows denote the lead compound (**1**, ChemDiv 7241-4207). The images were scanned at 532 nm (green, for reference spots) and 635 nm (red, for antibody signal). (**C**) Background subtracted average signal in arbitrary unit (A.U.) in the Cy5 fluorescent channel picked up by Drosophila (left) or human (right) TFIID is plotted against their respective BSA controls. Yellow circles depict the data points of the lead compound.

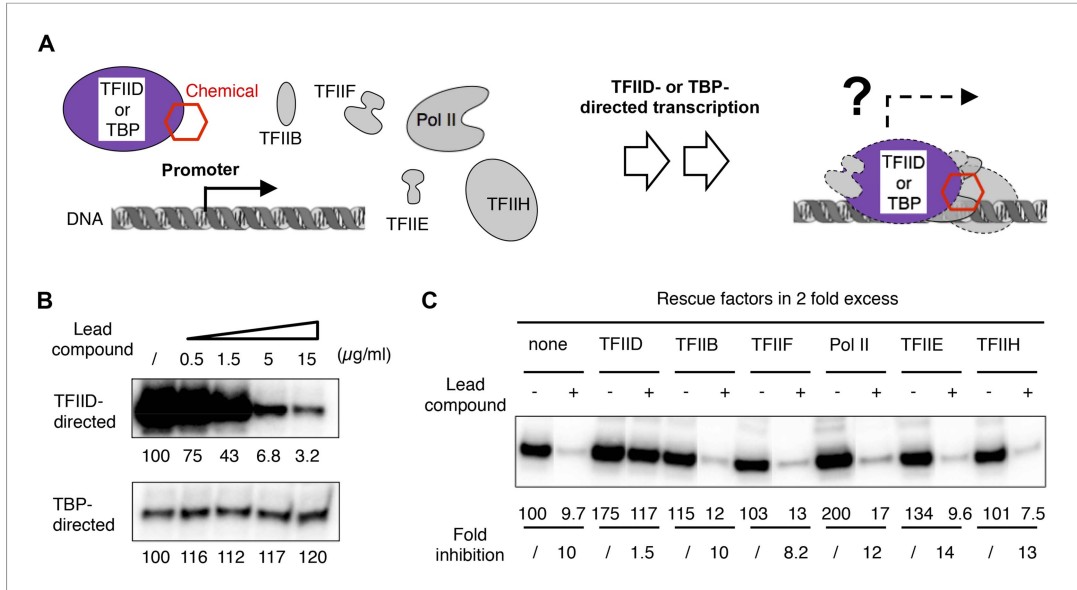

**Figure 2**. TFIID-specific transcription inhibition in a reconstituted system. (**A**) Cartoon illustration of TFIID- or TATA-binding protein (TBP)-directed transcription assays. Highly purified protein factors were mixed with the chemical and incubated before the addition of DNA templates for preinitiation complex (PIC) assembly. The DNA template contains the synthetic super core promoter (SCP1) (*Juven-Gershon et al., 2006*). (**B**) Dose-dependent inhibition of hTFIID-directed transcription, but not TBP-directed transcription, by the originally purchased lead compound (**1**, ChemDiv 7241-4207). The images were the primer extension products of the synthesized RNA and their signals were quantified and normalized to the respective controls (first lane from left, DMSO vector only). (**C**) Transcription rescue with individual protein factors supplemented in twofold excess (relative to the default dosage of each factor) immediately after chemical treatment (ChemDiv 7241-4207 at 5 µg/ml) and before the addition of the DNA template. Fold of inhibition was calculated for each reaction pair.

The following figure supplement is available for figure 2:

**Figure supplement 1**. TFIID dependency of the in vitro transcription assay and controls for the TFIID-specific inhibition.

## The inhibitor targets a histidine-rich IDR within TAF2

Identification of tin(IV) oxochloride led us to consider presumptive targets that should be histidine-rich domains of phylogenetically conserved TAF subunits. Testing this hypothesis, we found that a GST fusion of the histidine-rich *Drosophila* TAF2 (dTAF2) C-terminal fragment (residues [1125–1221]) bound selectively to the original tin-oxochloride containing sample in the arrayed library (*Figure 4A*). The targeted polypeptide fragment is part of a conserved IDR with adjacent low-complexity poly (K), poly (KH), and poly (KD/E) motifs found in both *Drosophila* and human TAF2 (*Verrijzer et al., 1994*; *Kaufmann et al., 1998*) (*Figure 4B* and *Figure 4—figure supplement 1*). The tin(IV) oxochloride cluster, with its hydrophilic, periodic surface features, presents a likely complementary ligand for these polar, repetitive, and histidine-rich IDRs.

To further validate tin(IV) oxochloride cluster as responsible for binding to the GST-dTAF2 (1125–1221) fragment, we carried out an independent surface plasmon resonance assay (*Figure 4C,D*). In this assay, the GST fusion protein was immobilized on a functionalized surface and an aqueous solution prepared with pure tin(IV) oxochloride chemicals was injected. We detected binding and dissociation of the chemical to the fusion protein upon chemical injection and buffer washing, respectively. In addition, we found that the binding is sensitive to imidazole and to pH values of ≤5.8, consistent with an essential role of histidine residues, which are protonated at lower pH and therefore incapable of coordinate bonding with the inhibitor. Similar results were observed in a parallel assay using a human TAF2 C-terminus (990–1199) fusion protein (*Figure 4E,F*), validating the conservation of the inhibitor target sites between *Drosophila* and human TAF2 proteins.

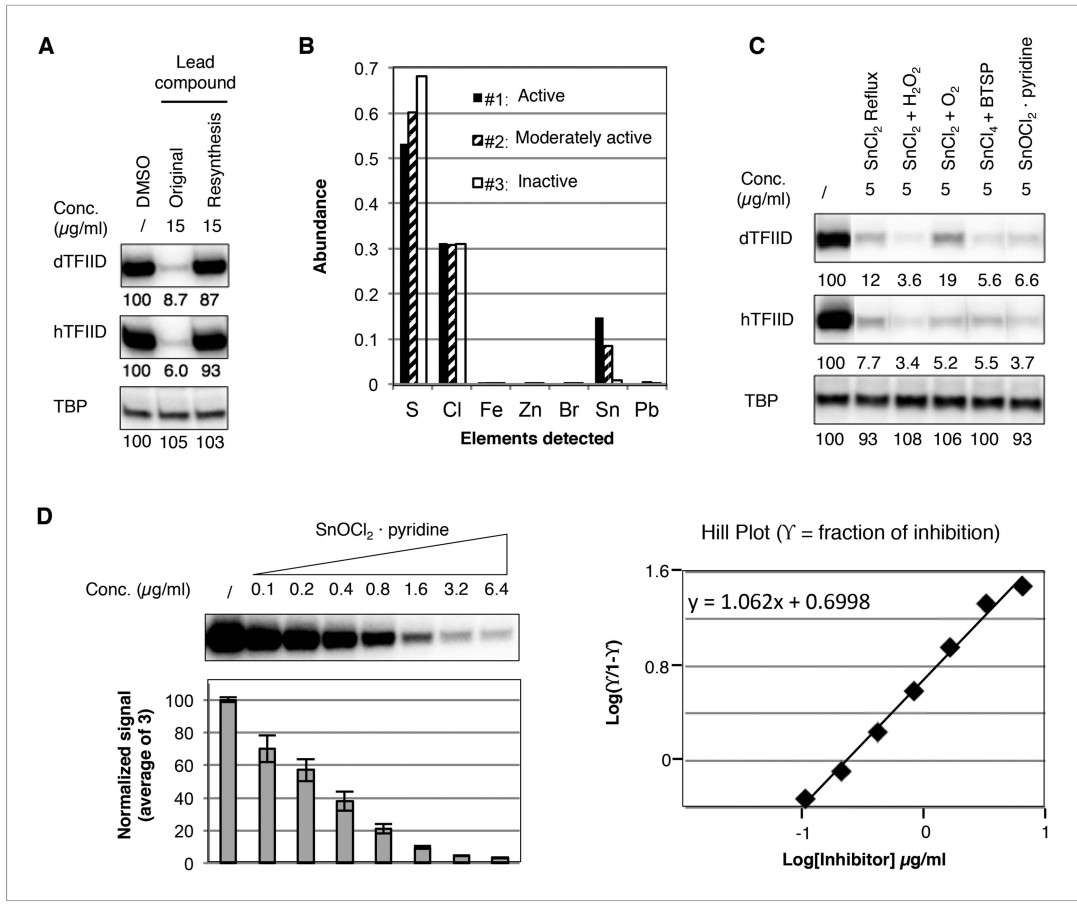

**Figure 3**. Identification of a tin(IV) oxochloride-derived cluster as the TFIID-specific transcription inhibitor. (**A**) Inhibition of TFIID-directed transcription by the originally purchased lead compound (**1**, ChemDiv 7241-4207), but not the in-house resynthesized one. (**B**) Elemental analysis of three batches of analog compound (**2**) with varying levels of inhibitory activity (see *Figure 3—figure supplement 1B,C*). (**C**) Inhibition of TFIID-directed transcription by tin(IV) oxochloride synthesized using different methods. From left to right: DMSO control, $SnCl_2$ refluxed in isopropanol, $SnCl_2$ oxygenation with $H_2O_2$ or $O_2$, and $SnCl_4$ oxygenation with *bis*(trimethylsilyl) peroxide (BTSP), and $SnOCl_2$ in complex with pyridine. (**D**) Dose response titration (left) and Hill Plot (right) of $SnOCl_2$·pyridine inhibiting hTFIID-directed transcription. Three independent replicates were used for plotting. The Hill coefficient was 1.062.

The following figure supplements are available for figure 3:

**Figure supplement 1**. Discrepancy between organic compound structures and the transcription inhibitor activity of commercial compounds.

**Figure supplement 2**. Tracking of the TFIID inhibitory activity to a tin-containing complex.

**Figure supplement 3**. Tin(IV) oxochloride-derived cluster identified as the TFIID-specific transcription inhibitor.

## The inhibitor may enhance TFIID binding to promoter DNA

Previous reports indicated that TAF2 can directly interact with DNA and is required for the recognition of the Inr element by TFIID (*Verrijzer et al., 1994*; *Kaufmann et al., 1998*). In addition, the inhibitor-targeted dTAF2 IDRs are rich in lysine residues that may non-specifically interact with negatively charged DNA backbone. We therefore examined whether GST fusions containing dTAF2 (1125–1221) or its sub-regions could retain DNA in a GST-pull down assay using DNA fragments corresponding to different regions of the promoter used in the transcription assay (*Figure 5A*). We found that indeed, this fragment of TAF2 can directly interact with DNA, primarily through the poly (K) and poly (KH)

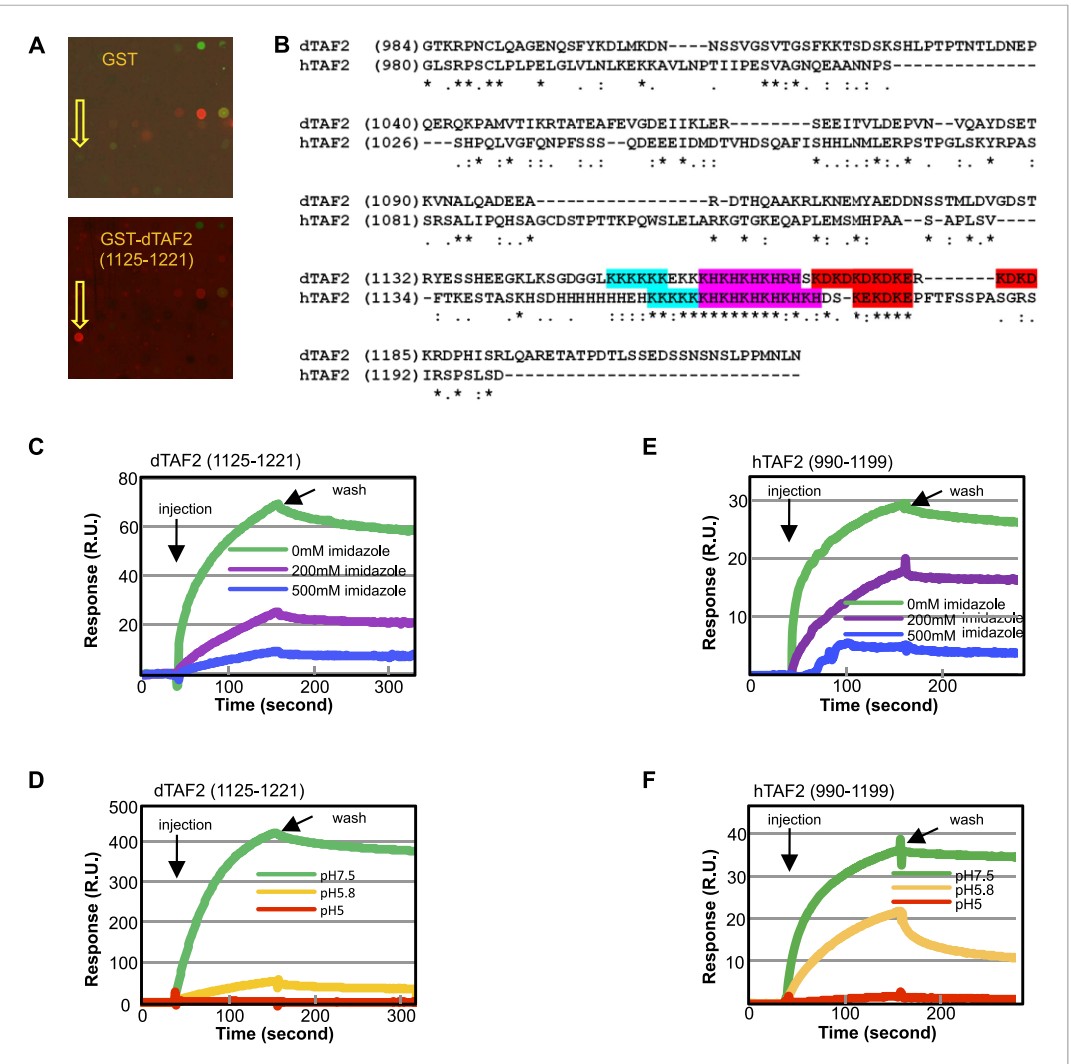

**Figure 4**. The inhibitor targets an IDR of TAF2 through histidines. (**A**) Binding of GST-dTAF2 (1125–1221) to the lead compound **1** (ChemDiv 7241-4207) (yellow arrow head) in the microarray. GST antibody was used for detection. Recombinant GST protein was the control. (**B**) Alignment of dTAF2 and hTAF2 C-terminal IDRs, with the conserved repetitive motifs highlighted. (**C**–**F**) Surface plasmon resonance sensorgrams of synthetic tin(IV) oxochloride cluster binding to GST-dTAF2 (1125–1221) (**C**, **D**) or Halo-hTAF2 (990–1199) (**E**, **F**) fragment, under varying imidazole concentrations (**C**, **E**) or pH (**D**, **F**) (R.U.: resonance unit).

The following figure supplement is available for figure 4:

**Figure supplement 1**. The intrinsically disordered nature of metazoan TAF2 C-terminus and the stringent conservation of and around the low-complexity sequences.

motifs, and more weakly through the adjacent poly (KD/E) motif. In addition, this interaction seems to be independent of the DNA sequence, which is not surprising given the low complexity of this region (*Figure 5A*). This suggests that this newly identified DNA interaction surface of TAF2 may be quite distinct from the previously reported Inr-selective DNA-binding activity of TAF2 (*Verrijzer et al., 1994*).

These results led us to re-examine the finding in *Figure 4A*, where the printed lead compound was bound by a bacterially expressed recombinant GST fusion protein (which likely contained co-purifying contaminating nucleic acids). We indeed found that interaction of GST-dTAF2 (1125–1221) with the tin-complex in the arrayed library was dependent on nucleic acids, as binding was ablated by nuclease

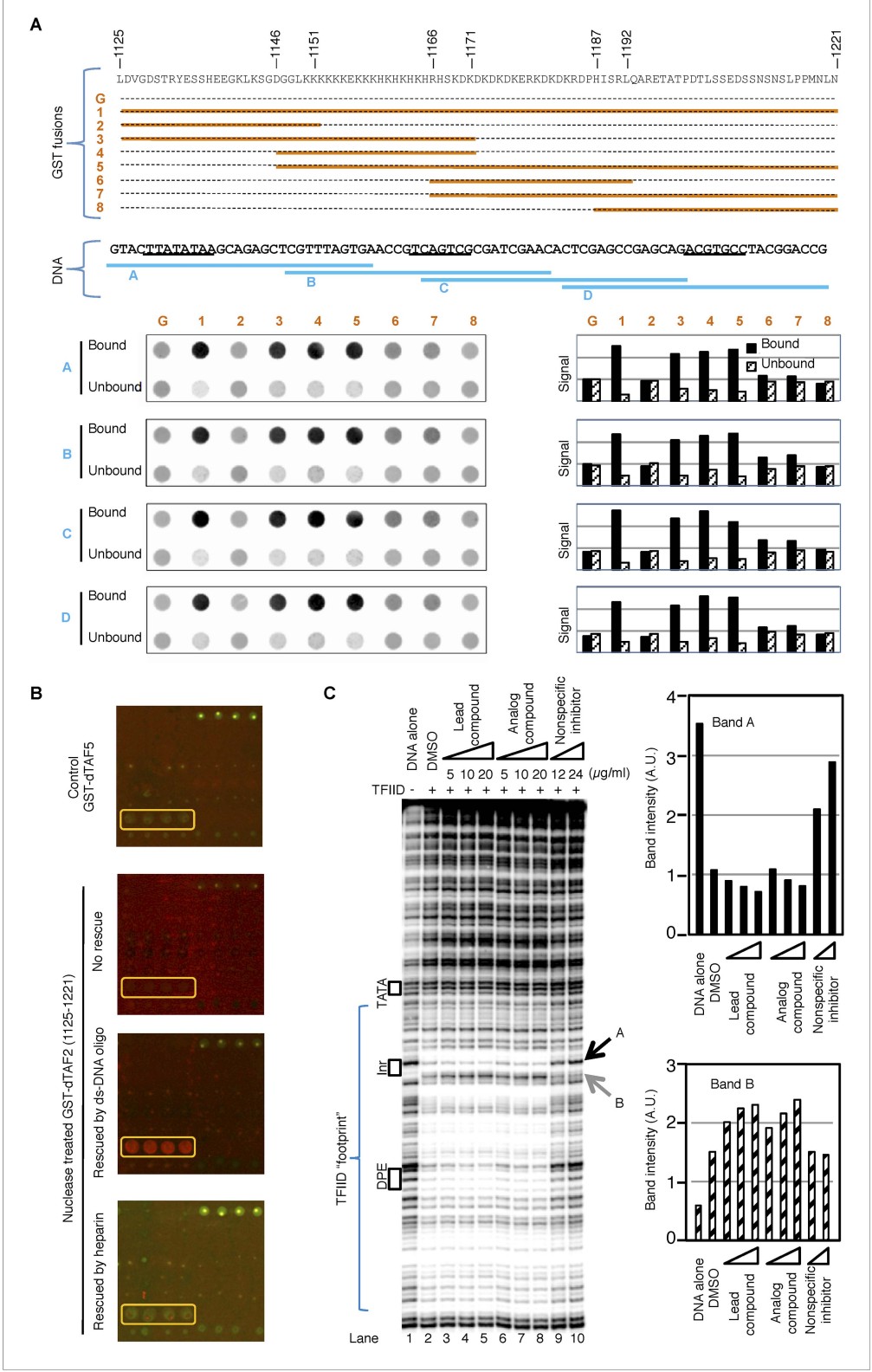

**Figure 5**. The inhibitor and DNA synergistically bind to the TAF2 IDR. (**A**) GST pull-down of double-stranded (ds) DNA oligonucleotides by GST-dTAF2 fragments. Top, the GST fusions (G: GST alone). Middle, the four DNA fragments tested, which are parts of the SCP1 (*Juven-Gershon et al., 2006*) used in this study (sequence shown, with the TATA, Inr, and DPE elements, from left to right, underscored). Bottom, DNA staining raw images (left) and
*Figure 5. continued on next page*

*Figure 5. Continued*

bar representation of the bound/unbound DNA signals (right). (**B**) Binding of nuclease-treated GST-dTAF2 (1125–1221) to the lead compound **1** (ChemDiv 7241-4207) printed in quadruplicates in the microarray (yellow rectangle), and its rescue by a double-stranded (ds)DNA oligonucleotide (1 µg/ml) or heparin (2 µg/ml). GST antibody was used for detection. Recombinant GST-dTAF5 was used as a negative control. (**C**) DNase I footprinting assay on TFIID-promoter binding, in the presence of the lead compound (**1**, ChemDiv 7241-4207), a structural analog (**2**, Princeton OSSK_462080), or an unrelated, non-specific (NS) inhibitor (Maybridge BTB08547, see *Figure 5—figure supplement 1B*). Shown is the digestion product of the end-labeled DNA template separated by gel electrophoresis. The DNA template contains the SCP1. Black boxes depict the positions of the TATA, Inr, and DPE elements, respectively. Blue bracket indicates the 'footprint' of TFIID. For simplicity, only two bands (denoted by arrowheads), which were protected (**A**) or intensified (**B**) upon TFIID binding, were selected for quantification (right).

The following figure supplement is available for figure 5:

**Figure supplement 1**. (**A**) DNase I footprinting assay of TFIID-promoter binding and its enhancement by the specific inhibitor at different salt concentrations.

pre-treatment and restored by addition of double-stranded DNA oligonucleotides (*Figure 5B*). Heparin, a poly-anionic mimic of DNA, did not rescue the binding, indicating that structural features of DNA were required, not simply its high density of negative charges (*Figure 5B*). These data suggested that DNA may stabilize an otherwise transient structural state of this IDR domain that then becomes susceptible to targeted binding by the tin compound.

We thus hypothesized that there is some sort of synergy between DNA and the inhibitor in binding to the TAF2-disordered region, and further examined this interaction in the context of holo-TFIID complex using a DNase I footprinting assay. We found the holo-TFIID protects an extended area of the promoter DNA from DNase I digestion, as previously reported on the super core promoter (SCP1) (*Juven-Gershon et al., 2006*) (*Figure 5C*). The signatures of this footprint include the protection of some hypersensitive sites (such as the band A at the upstream edge of the Inr element, and many bands covered by the bracket), and the exposure of new hypersensitive sites (such as band B at the downstream edge of the Inr element) (*Figure 5C*, lane 1 and 2). The TATA box is not protected because the engagement of TBP in the context of holo-TFIID complex at this promoter requires TFIIA (*Cianfrocco et al., 2013*). Interestingly, the inhibitor and its analog specifically enhanced features corresponding to promoter DNA binding by TFIID (lanes 3–8) (exemplified by the enhanced protection of band A and sensitization of band B, quantified by image analysis). This observed enhancement was subtle but highly reproducible using the various commercially supplied compounds and likely to be a specific consequence because a non-specific inhibitor picked up in our initial screen clearly reduced rather than enhanced TFIID binding (*Figure 5C* lanes 9–10 and *Figure 5—figure supplement 1*). Therefore, we speculate that the specific tin(IV) oxochloride chemical inhibitor may stabilize TFIID-promoter interactions under certain conditions via the TAF2 IDR.

## The inhibitor blocks Pol II engagement during de novo transcription initiation

It has generally been accepted that TFIID-promoter binding is a rate-limiting step in transcription initiation, and overall affinity of TFIID for promoter DNA correlates with transcription activity (*Juven-Gershon et al., 2006*). It is important to note that the inhibitor doesn't block TFIID-promoter binding, thus, it is likely to affect some downstream events. Individual TFIID-promoter contacts are likely dynamic, and a rate-limiting isomerization of TFIID could influence the transition from an initially bound Pol II to a productively engaged Pol II in the assembly of a functional PIC (*Yakovchuk et al., 2010*). We therefore hypothesized that an unnatural stabilization of the TAF2-DNA interaction by the inhibitor might specifically interfere with the presumptive conformational rearrangement required for the transition to a productively engaged Pol II.

To test this hypothesis, we first examined whether the inhibitor blocks transcription at the initiation stage. The primer extension assay used for transcription detection requires the synthesis of transcripts of 155 nucleosides, which could be limited by elongation. To identify the step(s) during transcription targeted by the inhibitor, we examined the synthesis of the very first dinucleotide during initiation,

using only adenosine triphossphate (ATP, the first nucleotide) and alpha-$^{32}$P labeled guanosine triphosphate (GTP, the second nucleotide) in the reaction. In this assay, a productive PIC would lead to the synthesis an ApG dinucleotide that is detected by autoradiography. Indeed, we observed inhibition of ApG synthesis (*Figure 6A*), suggesting the inhibitor acts very early during a stage at or before the synthesis of the first phosphodiester bond, and thus, very likely some step during PIC assembly.

To test which step during PIC assembly is affected by the inhibitor, we next performed a series of experiments in which DNA templates were pre-incubated with only a subset of protein factors ('GTF set 1', with factors incrementally added according to the order of PIC assembly [*Roeder, 1996*]) to maintain PIC assembly at specific stages prior to the addition of the inhibitor. After inhibitor treatment for 15 min, the missing factors were added (as 'GTF set 2') to complete PIC assembly and initiate transcription (*Figure 6B*, and *Figure 6C* lanes 1–14). As controls, we had either no factor or all the

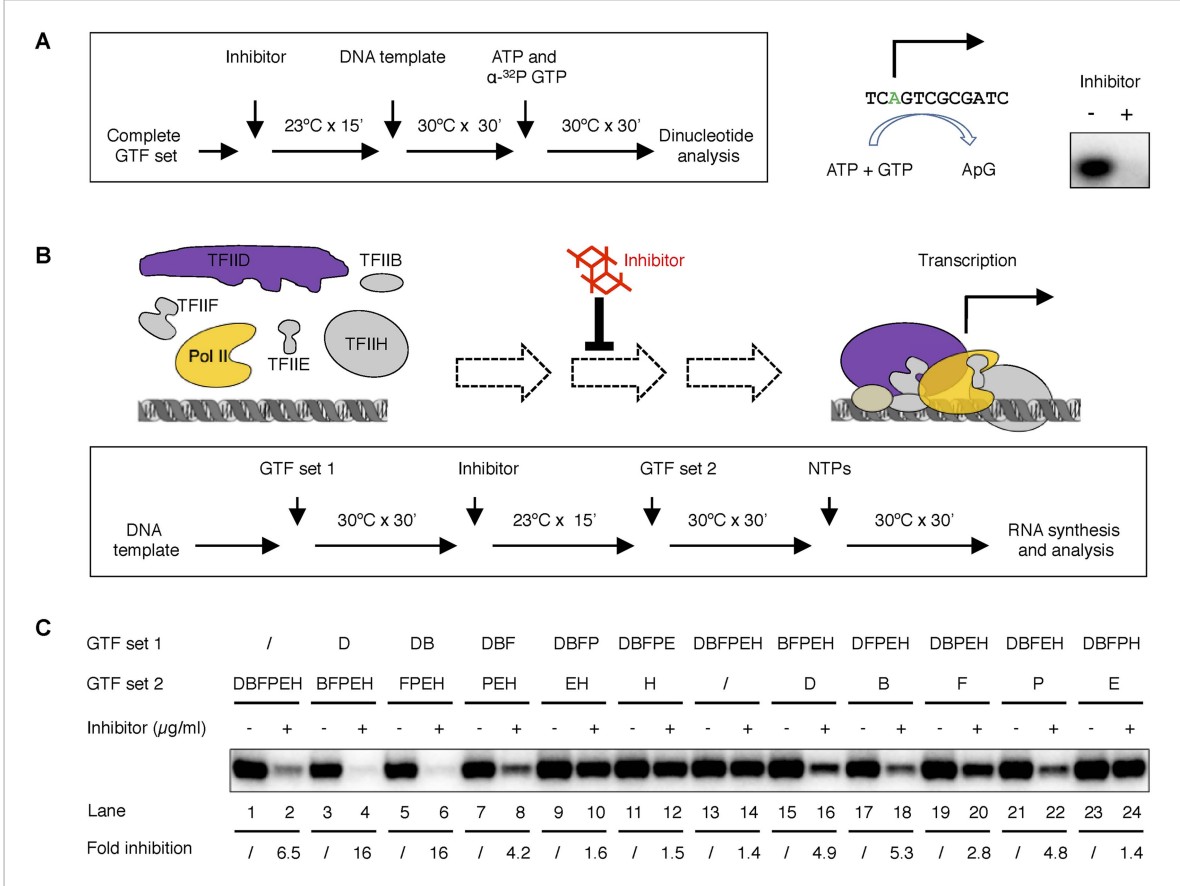

**Figure 6**. Tin(IV) oxochloride cluster specifically blocks de novo transcription initiation at the step of Pol II engagement. (**A**) Analysis of the first dinucleotide synthesis. Left, the scheme of the experiment. Green 'A' is where transcription starts. ATP and GTP are sufficient for the formation of the first phosphodiester bond at the SCP1. Right, the autoradiography image of the dinucleotide products. (**B**) Scheme of the step-wise PIC assembly perturbation experiment. The cartoon illustrates the question under investigation—which step of PIC assembly is inhibited. In box is the flow chart of the experiment. Inhibitor was added to the reaction after a subset of GTF (set 1) was incubated with the template DNA, followed by the addition of the rest of the protein factors (GTF set 2) for PIC assembly. All four nucleoside triphosphates (NTPs) were added in the end to allow RNA synthesis. (**C**) Gel image and quantification of transcription inhibition by treatment at different stages of PIC assembly. D, B, E, F, H, and P represent TFIID, TFIIB, TFIIE, TFIIF, TFIIH, and Pol II, respectively. '/' indicated the lack of any protein factor in GTF set 1 or 2. On the left part (lanes 1–14), GTF set 1 contains protein factors added incrementally one by one (from none to the complete set) following the order of in PIC assembly (*Roeder, 1996*). On the right part (lanes 11–14), individual factors were omitted in GTF set 1. SnOCl$_2$·pyridine was used at 5 µg/ml (**A**) or 2.5 µg/ml (**B**) as the inhibitor.

The following figure supplement is available for figure 6:

**Figure supplement 1**. The originally purchased lead compound specifically blocks de novo transcription initiation at the step of Pol II engagement.

requisite protein factors in GTF set 1 that would be equivalent to inhibitor treatment before the beginning or after the completion of PIC assembly (lanes 1–2 and 13–14). As expected, inhibitor treatment before PIC assembly efficiently blocked transcription (by 6.5-fold at 2.5 μg/ml, comparing lanes 1 and 2). In contrast, the inhibitor failed to interfere with transcription if added after PIC assembly (compare lanes 13 and 14), indicating some step(s) during initial PIC assembly is sensitive to the inhibition.

Interestingly, pre-incubation of TFIID or TFIID together with TFIIB with the DNA template led to a more severe inhibition (16-fold, *Figure 6C*, lanes 4 and 6) when compared to adding all the protein factors after the inhibitor (6.5-fold, lane 2). This suggests that a TFIID-TFIIB-DNA sub-complex is likely the most susceptible substrate for inhibition, while the presence of other core promoter factors may facilitate the progression of PIC assembly and reduce the time interval of this vulnerable stage. Addition of TFIIF partially alleviated the inhibition (comparing lanes 6 and 8), suggesting that TFIIF may play a necessary but insufficient role in driving TFIID into an inhibitor-resistant state. Upon further addition of Pol II to the assembly, the system became resistant to the inhibitor to a level comparable to that of preformed PICs (lanes 10 and 14). In contrast to the essential role of TFIID, TFIIB, TFIIF, and Pol II (the same set of factors required for Pol II engagement) in forming the minimal PIC intermediate resistant to inhibition, adding TFIIE and TFIIH had no effect on inhibition (comparing lane 10, 12, and 14), even though these two factors are required for robust transcription. Very similar results were observed when individual protein factors were left out one by one from GTF set 1 as an alternative strategy to arrest PIC assembly at specific stages (*Figure 6C* lanes 11–24 for the pure tin(IV) oxochloride compound; and *Figure 6—figure supplement 1* for the original lead compound ChemDiv 7241-4207). Taken together, these results strongly suggest that the step most sensitive to inhibition involves the initial binding of Pol II to a DNA-TFIID complex that must then transition into a conformation compatible with productive Pol II engagement. Importantly, after Pol II engagement takes place during de novo PIC assembly, the system becomes resistant to the inhibitor.

## The inhibitor arrests an isomerization step required for full Pol II engagement

To better understand how Pol II engagement might be inhibited by the chemical, we performed DNase I footprinting assays to directly examine the potential conformational isomerization events during PIC assembly. In this assay, various PIC components were incubated with the template DNA under conditions that would lead to optimal transcription output if all the other components were included. We found that, as expected, TFIID alone caused a footprint covering the Inr, DPE, and extending downstream (to ~+55) (*Figure 7A*, lane 1 and 2). The addition of TFIIB enhanced protection over the upstream region (from the TATA box to the Inr), and this protection became further enhanced by the addition of TFIIF (lanes 3 and 4), consistent with the synergy between TBP and TFIIB binding to promoter DNA (*Tsai and Sigler, 2000*), and the stabilization of TFIIB binding by TFIIF (*Luse, 2012*). Interestingly, we observed significant changes in the footprint pattern upon the addition of Pol II (lane 5). These changes include (i) the exposure of some hypersensitive sites protected by the initial TFIID binding (such as position ~+16, and the sites flanking the DPE), suggesting the release or unmasking of some DNA from the bound TFIID; (ii) emergence of new hypersensitive sites (such as position ~+14), suggesting structural changes in the DNA trajectory itself caused by Pol II binding; (iii) reduction of the hypersensitive site induced by TFIID binding at the edge of the Inr, consistent with the release of some DNA from TFIID and/or the association of Pol II; and (iv) a strong and extended protection covering the upstream of the TATA box (~−37) to the Inr, consistent with the engagement of Pol II and other PIC components with this region of the promoter. These results are also consistent with a previous report using a different promoter (*Yakovchuk et al., 2010*) that suggested some kind of a conformational isomerization at the promoter associated with Pol II engagement.

To examine the effect of the chemical inhibitor on this isomerization step, we used suboptimal levels of TFIID to first assemble a TFIID-TFIIB sub-complex at the promoter. This sub-complex gives a weak footprint (that is expected to be more sensitive to perturbation than the footprint generated by optimal levels of TFIID) (*Figure 7B*, lanes 2 and 3). Despite using suboptimal levels of TFIID, no significant change in the footprint protection was observed upon inhibitor treatment (lane 4). This is consistent with the hypothesis that the inhibitor likely acts at a stage after TFIID and TFIIB binding. Because Pol II and TFIIF are known to be tightly associated to each other physically and functionally (*Roeder, 1996*) and they both rescue PIC assembly from the inhibition (TFIIF partially and Pol II

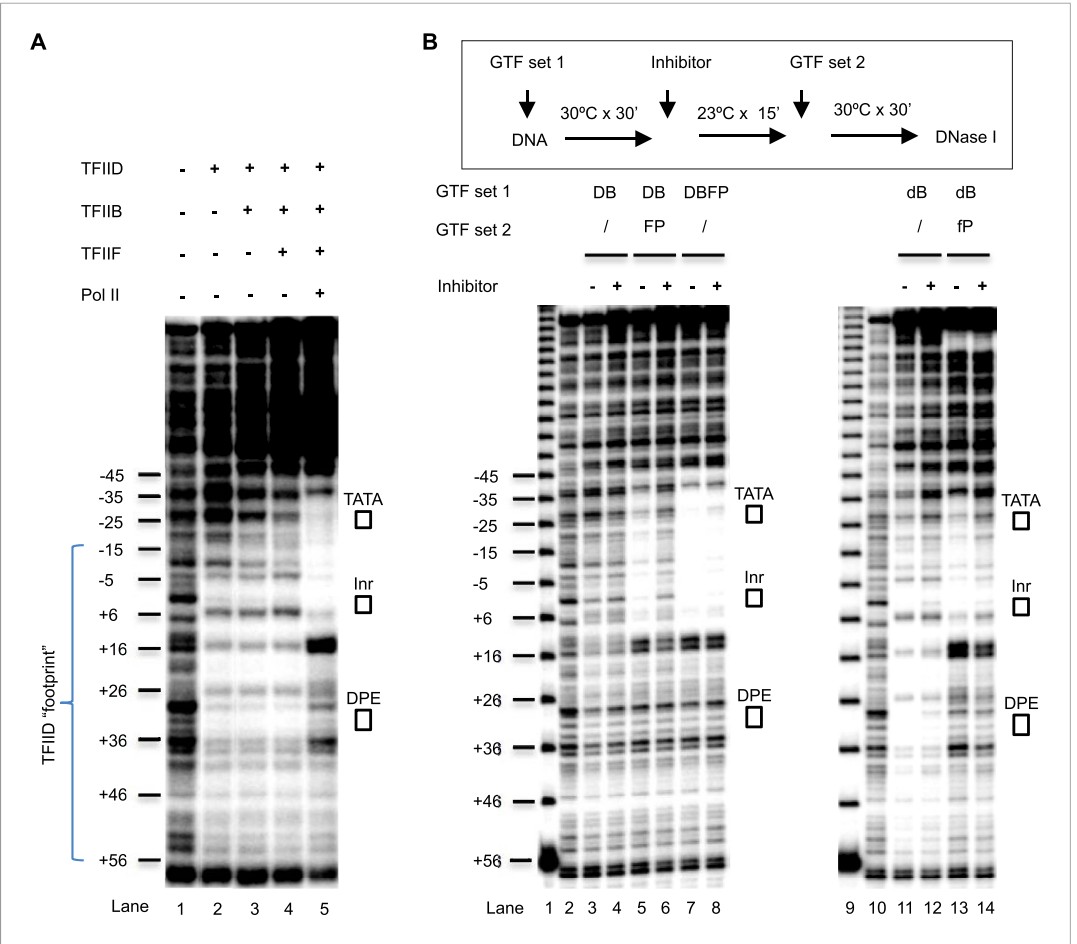

**Figure 7**. DNase I footprinting assays monitoring PIC assembly and structural isomerization. (**A**) Early steps of PIC assembly. Specified GTFs were incubated with the end-labeled DNA template, followed by DNase I digestion. Blue bracket highlights the TFIID footprint. The numbers are relative to the transcription start site (+1). (**B**) Arresting of the conformational isomerization. Top, the scheme. The inhibitor used here was SnOCl$_2$·pyridine. Lanes 1 and 9 are 10 bp DNA ladder. Lanes 2 and 10 are digestion of naked DNA. The lower case 'd' and 'f' in lanes #11–14 reflect the use of less TFIID and TFIIF (together with the omission of spermidine and carrier nucleic acid in the reaction—see 'Material and methods' for detail). Letter abbreviations are explained in *Figure 6* legend.

completely), we next added them together to the reaction and observed, as expected, the characteristic footprint changes associated with Pol II engagement (lane5). Importantly, in the presence of the inhibitor, the Pol II-induced footprint changes over the TATA box, the Inr, and in between were largely abolished, while the protection downstream of the Inr were partially altered, suggesting that the full engagement of Pol II is significantly disrupted by the tin compound along a major portion of the promoter (lane 6). In control experiments where the inhibitor was added after TFIID, TFIIB, TFIIF, and Pol II, it had almost no detectable effect on the footprint corresponding to the post-isomerization state of the partial PIC (lanes 7 and 8). This is consistent with the transcription profile where addition of Pol II renders the system resistant to the inhibitor (*Figure 6C*). To better focus on the changes over the downstream promoter region, we also performed the experiment under conditions that generate optimal TFIID footprints. We verified that the inhibitor only partially compromised the downstream changes associated with Pol II engagement, although it almost completely blocked the changes covering the TATA and Inr elements (lanes 9–14). The changes of DNase I protection patterns observed in the downstream region suggest that the inhibitor may still allow Pol II to partially interact with this part of the promoter (either directly or indirectly via TFIID), and that a simple complete exclusion of Pol II from accessing promoter DNA is unlikely to be the

primary inhibition mechanism. These results are consistent with the possibility that the inhibitor directly stabilizes TFIID–Inr interactions and prevents the progression of the conformational isomerization. We conclude that transcription inhibition is most likely a consequence of incomplete Pol II engagement due to arrested TFIID isomerization. Once PIC assembly passes through this Pol II-dependent isomerization stage, the inhibitor apparently can no longer interfere with transcription.

## Re-entry of Pol II during reinitiation is resistant to the inhibition

The tight correlation between Pol II engagement and inhibitor sensitivity does not rule out the possibility of a mutual steric exclusion between inhibitor binding to the TAF2 IDR and Pol II engagement at the promoter, which may be independent of the conformational isomerization. Because Pol II and TFIIF are thought to leave the promoter during the elongation phase after de novo transcription initiation, while TFIID may be retained at the promoter as part of a reinitiation scaffold (*Zawel et al., 1995*), we reasoned that examination of transcription reinitiation in our system may help to further dissect the inhibition mechanism.

A specialized transcription reinitiation process that is distinct from the process of de novo PIC assembly pathway has been reported for in vitro transcription systems supported by human and yeast nuclear extracts or purified factors (*Hawley and Roeder, 1987*; *Zawel et al., 1995*; *Yudkovsky et al., 2000*). Although TFIID is thought to be part of the reinitiation scaffold, if and how the retained TFIID might differ from complexes assembled during de novo initiation has remained unclear. In addition, using *Drosophila* embryo nuclear extracts and a dual template assay system, Kadonaga has demonstrated that each round of transcription requires complete assembly and disassembly of the PIC. Thus, at least in vitro, under certain transcription conditions, there is no commitment of any limiting transcription factor to the initial template to form a pre-licensed reinitiation scaffold (*Kadonaga, 1990*). This observed difference is likely a result of various regulatory protein factors and/or promoter elements present in different reaction systems. We therefore decided to further investigate whether TFIID is committed to an initiating DNA template and becomes pre-licensed for reinitiation in our completely defined highly purified human Pol II transcription system.

We first performed transcription reactions with two DNA templates that produce transcripts of distinct lengths (*Figure 8A*). We incubated the first DNA template (DNA 1) with a complete set of protein factors to form a stable PIC, then added the second DNA template (DNA 2) to see whether the essential factors are still available. To constrain transcription to a single round so that we can compare de novo PIC assembly on the two templates, we added 0.1% Sarkosyl immediately after the addition of nucleoside triphosphates (NTPs) for RNA synthesis following previously reported procedures using crude nuclear extracts (*Hawley and Roeder, 1985*; *Kadonaga, 1990*), which is also validated in our highly purified system (*Figure 8—figure supplement 1A*). We found that the presence and pre-incubation of DNA 1 severely compromised (by 11–18-fold) transcription from DNA 2 (*Figure 8A*, compare lanes 2–4 with 7 for the short transcript, and lanes 9–11 with 14 for the long transcript), indicating that some limiting factors become stably committed to the first DNA template during initial PIC assembly. Adding more fresh TFIID (1 × equivalent to the initial dosage) immediately after DNA 2 restored transcription activity from DNA 2 by ~fourfold (comparing lanes 4 with 5, and 11 with 12), suggesting that TFIID is likely one of the limiting and DNA template committed factors during de novo PIC assembly.

Omitting Sarkosyl treatment so that transcription can go through multiple rounds led to an increase in transcription signal by ~threefold (comparing lanes 1, 4, and 6 for the long transcript, and lanes 8, 11, and 13 for the short transcript), indicating the detection of one de novo round of transcription initiation plus ~2 rounds of reinitiation. In the absence of Sarkosyl treatment, the presence and pre-incubation of DNA 1 with the protein factors still severely compromised (by 12–16-folds) transcription from DNA 2 (comparing lanes 15 and 17 for the short transcript, and lanes 18 and 20 for the long transcript). As expected, this template commitment under multi-round transcription conditions can also be partially alleviated by the addition of extra TFIID (by ~sixfold) (lanes 16 and 19). These findings suggest that commitment of the limiting factor (including TFIID) to DNA 1 during de novo transcription initiation mostly persists through multiple rounds of reinitiation. Therefore, as reported with other human transcription systems, our highly purified system also involves a stable reinitiation scaffold of which TFIID is likely a critical component.

To probe the functional states of TFIID (particularly that of the TAF2 IDR) during transcription reinitiation, we treated the reaction with the inhibitor either before or after PIC assembly, either under

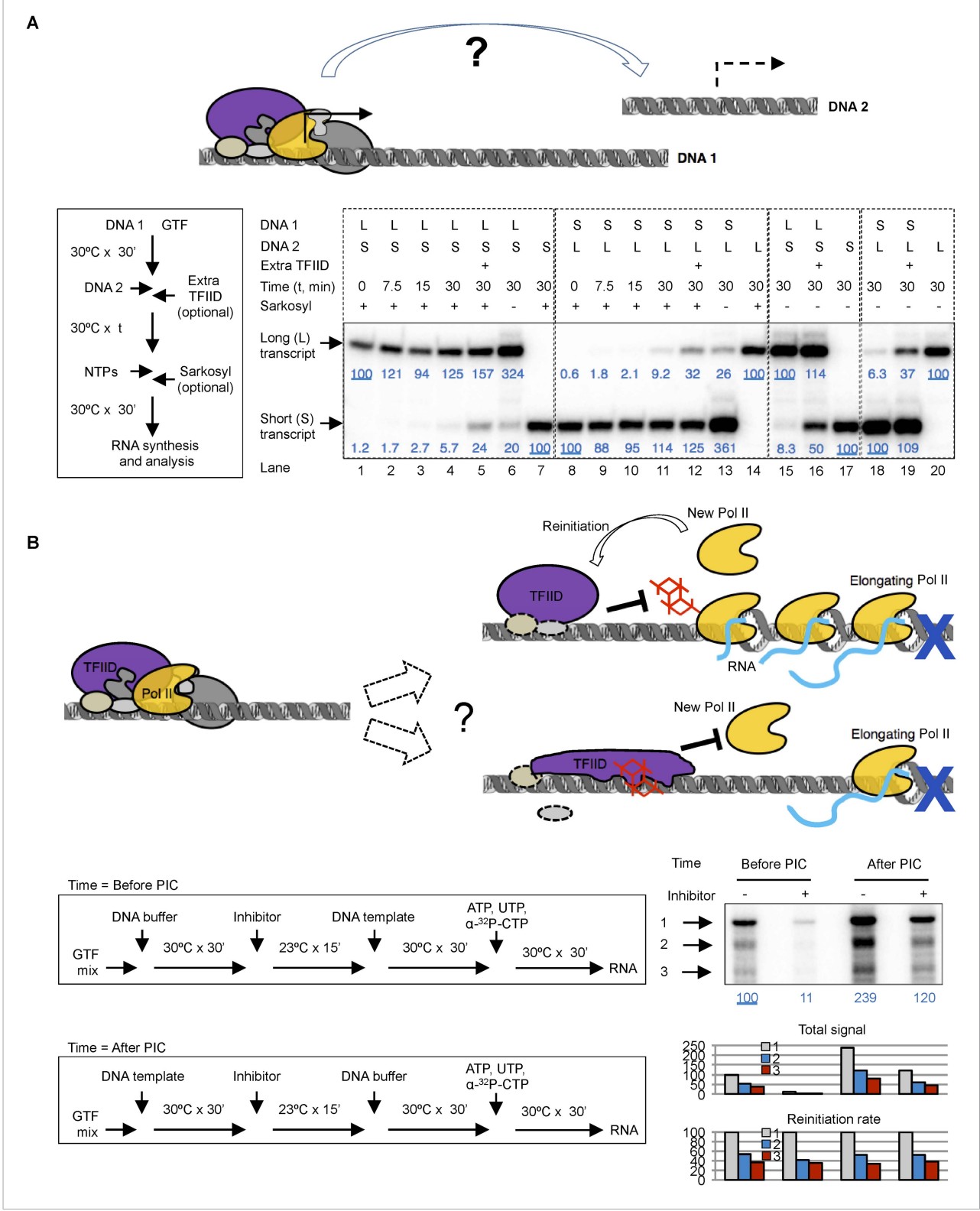

**Figure 8**. Template commitment of TFIID and its resistant to tin(IV) oxochloride inhibition during reinitiation. (**A**) Two-template assay to test the template commitment of GTFs. The cartoon illustrates the question. Bottom left is the experimental scheme. After PIC assembled on the first DNA template (DNA 1), the second DNA template (DNA 2) was added (as an option, onefold extra TFIID can also be added immediately following DNA 2) and incubated for specified time, followed by the addition of nucleoside triphosphates (NTPs) for RNA synthesis. Sarkosyl was added to a final concentration of 0.1%

*Figure 8. continued on next page*

*Figure 8. Continued*

(within 30 s after the addition of NTPs) as an option to restrict transcription to a single-round. The two DNA templates both contain a SCP1, but lead to primer extension products of different length (L: long, 192 bases; S: short, 155 bases). Bottom right is the results, sub-divided into four groups (dashed boxes). The transcription signals were normalized within each group for each (L or S) specific primer extension product (shown in blue immediately under the specific bands). (**B**) Comparison of single-round transcription vs multiple-round transcription. The cartoon illustrates the question to address: whether TFIID in the reinitiation scaffold is sensitive to inhibition or not. Blue 'x' indicates the position of the first G residue where the first Pol II will be stalled in the absence of GTP. Bottom left is the scheme. 'DNA buffer' contained no template DNA. SnOCl$_2$·pyridine was used as the inhibitor. Bottom right is the result. Black arrows indicate the bands corresponding to the first (1), second (2), and third (3) transcript synthesized from the DNA template. Blue numbers are the normalized quantification of the first transcript from each lane. All three band intensity is plotted as 'Total signal' (the first transcript of the first lane from the left was arbitrarily set as 100). Reinitiation rate was plotted by setting the first transcript of each lane as 100 (to calculate the chance of the second and third round of transcription to occur in each reaction).

The following figure supplement is available for figure 8:

**Figure supplement 1**. Controls for reinitiation experiments.

the condition that would allow multiple rounds of transcription or under Sarkosyl treatment condition limiting transcription to the first, de novo round (*Figure 8—figure supplement 1B*). As expected, tin-compound treatment before PIC assembly efficiently inhibited TFIID-directed transcription (comparing lanes 3, 4, 7, and 8 from left) under both conditions. On the other hand, when the inhibitor was added after PIC assembly (before the addition of NTPs, which in turn is before the formation of the reinitiation scaffold), there was no inhibition of transcription under either the single (de novo initiation) or multiple (de novo plus reinitiation) round conditions, indicating that reinitiation and the re-entry of Pol II to the reinitiation scaffold is largely refractory to inhibition by the tin-oxochloride compound.

To further independently validate the resistance of the reinitiation scaffold to inhibition, we used a colliding polymerase assay (*Szentirmay and Sawadogo, 1994*). In this assay, a sequence containing no G residues was inserted after the adenovirus major late promoter (for this 'G-less' cassette-based assay, the SCP1 is not suitable because it contains multiple G residues in the initially transcribed sequence). Transcription in the absence of GTP (but in the presence of all the other three required nucleoside triphosphates) would cause the first Pol II molecule to stall at the position of the first G-residue after the G-less cassette, and to stall the second and third Pol II molecules at positions further upstream. Using alpha-$^{32}$P labeled CTP as a substrate, this system will allow the first Pol II molecule to synthesize a nascent transcript corresponding to the length of the G-less cassette, followed by progressively shorter transcripts that result from reinitiation (*Szentirmay and Sawadogo, 1994*). Using our highly purified transcription factors, we observed just such a ladder of nascent transcripts of expected sizes as previously reported (*Szentirmay and Sawadogo, 1994*). Importantly, addition of 0.02% Sarkosyl just before the NTP substrates only weakly diminished the synthesis of the longest transcript, but almost completely abolished the shorter ones (*Figure 8—figure supplement 1C*), mirroring the effect of 0.02% Sarkosyl in mildly affecting preassembled PIC for the first round of transcription while completely blocking reinitiation (*Figure 8—figure supplement 1A*). Thus, the longest transcript likely represents the products of de novo (first round) transcription, while the short ones represent reinitiation products in our system. Quantification of the longest transcription confirmed robust (ninefold) inhibition of de novo transcription initiation when the chemical was added before PIC assembly, and this inhibition became highly attenuated (twofold) when the chemical was added after PIC assembly. When we normalized the signal from each round of transcription in each reaction to the first round of transcription, we found the ratio of the second and third transcripts to the first one ('reinitiation rate') to be highly consistent across all the samples. In particular, inhibitor added after PIC assembly (but before NTPs allow the first Pol II to escape the promoter) had no effect on reinitiation rate, confirming that the chance of a reinitiation scaffold to support more transcription events (after the first Pol II escapes the promoter) is not sensitive to the inhibitor.

Taken together, our studies on reinitiation suggest that after Pol II leaves the promoter, the template-committed TFIID complex remains in an inhibitor-resistant state, distinct from that of the TFIID-TFIIB-DNA intermediate during the early steps of de novo initiation that appears to be the target of the inhibitor. Therefore, the inhibitor-resistance caused by Pol II engagement during de

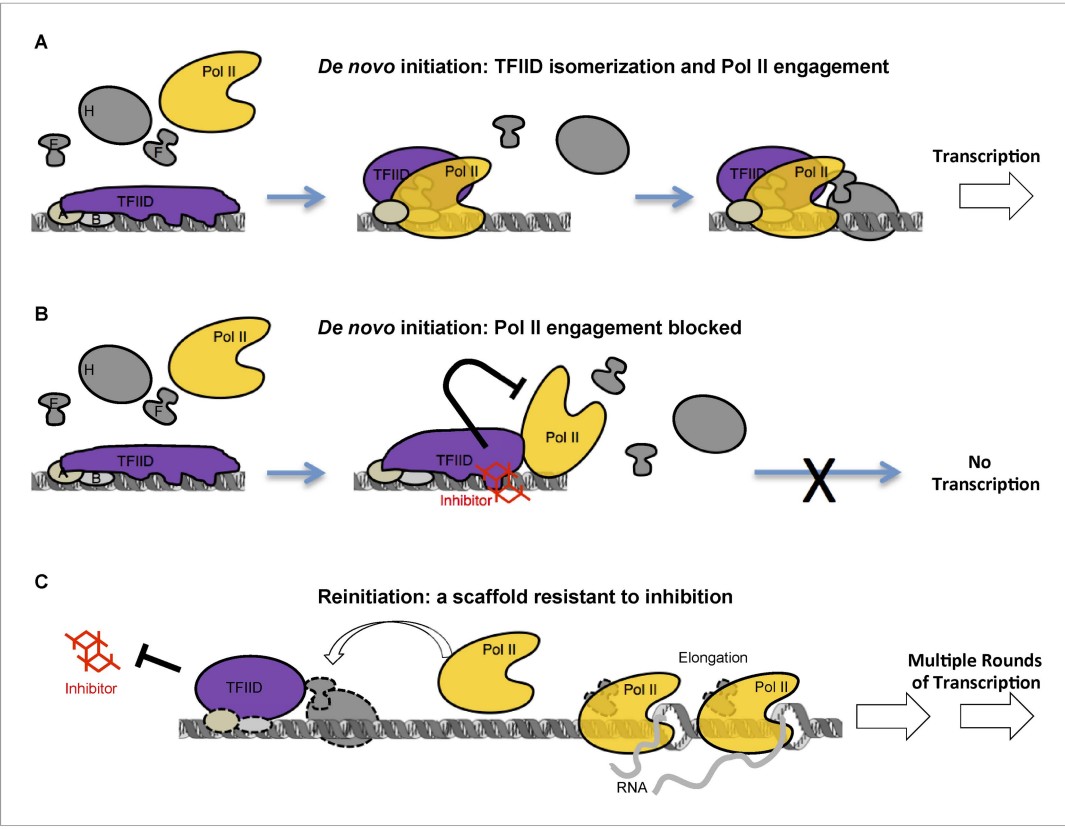

**Figure 9**. A model of inhibition that mechanistically distinguishes the two modes of transcription initiation. (**A**) Initially, TFIID forms multiple contacts with an extended promoter DNA region that is stabilized by TFIIA and TFIIB. TFIIA doesn't affect transcription at this promoter with purified factors, but it does facilitate the TATA box protection by TFIID alone (*Cianfrocco et al., 2013*) or by TFIID together with TFIIB (ZZ and RT unpublished). We propose a critical isomerization step during de novo PIC assembly involving a TFIID conformational change (i.e., release of at least part of the promoter DNA, illustrated by the change in the shape of TFIID) to allow entry and engagement of Pol II. Once Pol II becomes engaged and further stabilized by other factors (TFIIE, TFIIF, etc) transcription can proceed. (**B**) The inhibitor, by binding and interfering with the TAF2 IDR, arrests TFIID isomerization and Pol II engagement, thus, blocking the assembly of a functional PIC. DNase I footprint assay reveals that Pol II molecules can still partially interact with the downstream portion of promoter DNA in the presence of the inhibitor. (**C**) Once the first round of Pol II engagement is accomplished and isomerization has occurred, the PIC intermediate establishes a state resistant to inhibition. After Pol II enters the elongation phase, TFIID remains at the isomerized state as part of a reinitiation scaffold. This reinitiation complex bypasses the initial stages of de novo PIC assembly where TFIID contacts an extended DNA region and thus is resistant to the inhibition by the tin(IV) oxochloride cluster. In addition, this shortcut may be a mechanism for the reinitiation scaffold to facilitate reloading of more Pol II molecules.

novo PIC assembly is unlikely simple steric masking, but rather a result of rearrangement of TFIID–DNA interactions. This difference in TFIID conformation and functional state between de novo initiation and reinitiation also suggests how TFIID and the pre-licensed scaffold might bypass certain stages of de novo PIC assembly to facilitate reinitiation. Accordingly, a model is proposed summarizing the mechanistic insights revealed by the inhibitor (*Figure 9*).

## Discussion

In this study, we identified an unconventional chemical inhibitor of Pol II transcription and used it to probe the function of an IDR within the TAF2 subunit of metazoan TFIID. Although IDRs are abundant, mechanistic studies of their functions have been challenging. The TAF2 IDR targeted by the tin(IV) oxochloride-derived inhibitor turned out to be a newly identified TAF2 DNA-binding domain that likely becomes structured upon DNA binding, perhaps reminiscent of basic domains in the leucine

zipper family of activators (*Shuman et al., 1990*; *Weiss et al., 1990*). On the other hand, the TAF2 IDR domain, with its low-complexity sequences, does not appear to be responsible for the sequence-specific recognition of the Inr element by TAF2 (*Verrijzer et al., 1994*). Instead, this IDR domain apparently binds DNA non-specifically, consistent with our finding that template DNA bearing Inr mutations still responded to the inhibitor in transcription reactions.

One possible explanation for the inhibition mechanism is an over-stabilization of TAF2 IDR-DNA interaction by the inhibitor. In this model, the inhibitor-stabilized TAF2 IDR is an impediment to full Pol II engagement. We postulate that this Pol II-dependent isomerization is a multi-step process that engages an extended region of the promoter DNA that gets blocked at the Inr by the inhibitor. This model is consistent with our DNase I footprinting results. Alternatively, if the TAF2 IDR is an active helper for Pol II to overcome the barrier posed by other bound factors (like TFIID) to interact with the upstream DNA and the associated protein factors (such as TFIIB), interference of TAF2 IDR function could also lead to a partially impeded Pol II engagement. This is reasonable because IDRs are known to interact with multiple partners with diverse functions (*Uversky, 2013*). Based on the different levels of sensitivity to inhibition, Pol II engagement can be dissected into at least two sections: the release of the downstream elements from TFIID (which can partially occur in the presence of the inhibitor) and the binding and reorganization of the TATA-Inr region (which is almost completely inhibited). Further study using the TFIID inhibitor and advanced techniques, such as single-molecule imaging suitable to capture fast dynamics in complex biochemical reactions (*Revyakin et al., 2012*), could be useful to reveal additional mechanistic aspects of this elusive part of PIC assembly.

Given that IDRs are highly sensitive to their local molecular environment (*Uversky, 2013*), it is possible that the TAF2 IDR may be selectively targeted by some cellular factors to modulate transcription initiation in a manner analogous to the action of the tin-based inhibitor described here. Interestingly, although generally considered a coactivator (*Albright and Tjian, 2000*), TFIID has also been found at promoters of silenced/inactive genes (*Breiling et al., 2001*; *Tatarakis et al., 2008*) and may even function as a 'checkpoint' during activation (*Marr et al., 2006*). It remains to be seen whether the TAF2 IDR is required for such reported checkpoint functions. Our findings regarding the TAF2 IDR are reminiscent of regulatory functions mediated by the AT-hook domains of TAF1. Much like TAF2, the TAF1 subunit of TFIID also plays critical roles in core promoter recognition (*Juven-Gershon and Kadonaga, 2010*). The two AT-hook DNA-binding domains of TAF1 preferentially associate with the narrower minor grooves associated with AT-rich sequences (*Metcalf and Wassarman, 2006*). Most interestingly, AT-hooks are also intrinsically disordered (*Liu et al., 2006*), and their phosphorylation by casein kinase II was proposed to regulate core promoter selectivity by altering the conformation of TFIID (*Lewis et al., 2005*). Therefore, the various intrinsically disordered DNA-binding domains of TFIID subunits may underscore previously underappreciated versatility of this complex in recognizing core promoter elements and responding to activators.

The tin-based chemical reported here is the first known reagent capable of directly and mechanistically distinguishing the functional states of TFIID during initiation and reinitiation and discriminating between these two modes of transcription. Reinitiation is likely an important aspect of transcription regulation in vivo, possibly related to 'bursting' (*Raj et al., 2006*), but its mechanism is poorly understood. We found that although the engagement of Pol II during de novo PIC assembly is sensitive to the inhibitor, re-entry of Pol II during reinitiation is inhibitor resistant. The mechanism behind this difference is linked to the different functional states of TFIID and the TAF2 IDR targeted by the inhibitor. As a reagent for mechanistic studies, the tin(IV) oxochloride cluster is distinct from Sarkosyl, which has traditionally been used to study reinitiation (*Hawley and Roeder, 1987*). Sarkosyl in fact does not discriminate between initiation and reinitiation, even in cases where specialized reinitiation scaffolds are involved. Proper targeting of reinitiation by Sarkosyl relies on the timing of addition (after de novo PIC assembly but before reinitiation in synchronized reactions). In addition, the actual target of Sarkosyl has remained largely a mystery. In contrast, the tin(IV) oxochloride cluster identified here has a clear and specific target and obviates the necessity for timed addition by directly blocking initiation while leaving reinitiation and on-going transcription intact.

We should point out that de novo initiation and reinitiation may not necessarily represent mechanistically distinct pathways under certain conditions in vitro. Numerous possibilities have been proposed by researchers for the apparent differences in observing disengagement of PIC components at each round (*Kadonaga, 1990*) vs retention of 'reinitiation scaffolds' for reinitiation through pathways distinct from de novo transcription initiation (*Hawley and Roeder, 1987*; *Zawel et al., 1995*;

*Yudkovsky et al., 2000*). For example, the role of activators and/or specific promoters in retaining TFIID, Mot1 proteins in removing and recycling TFIID, and other limiting factors, etc has all been proposed to influence these two potential mechanistically distinct pathways to achieve multiple rounds of transcription. It is also possible that the reported discrepancies can be a reflection of the complexity and diversity of regulatory factors assembled as part of the core transcription machinery responsible for promoter recognition in different organisms and possibly even distinct cell types.

Although the newly identified compound may present complications for direct in vivo applications, this reagent revealed an intriguing possibility of specifically blocking de novo transcription initiation without affecting reinitiation via targeting of TFIID. Interestingly, the canonical TFIID complex has been reported in several cases to be dispensable for ongoing transcription in terminally differentiated, non-dividing cells (*Cler et al., 2009*; *Goodrich and Tjian, 2010*; *Muller et al., 2010*). Accordingly, these cells are likely to be resistant to perturbation by chemicals that specifically target the canonical TFIID complex, particularly the function of TFIID in de novo transcription initiation. In contrast, biological processes stringently dependent on de novo transcription initiation, such as the onset of viral transcription after infection or re-establishment of mRNA synthesis in rapidly dividing cancer cells, should be more sensitive to this kind of chemical perturbation. Therefore, with specific functions required for de novo transcription initiation but not reinitiation, the canonical TFIID complex has the potential to be even more selective than TFIIH, another component of the Pol II core transcription machinery that was recently found to be a promising target for cancer therapy development (*Titov et al., 2011*; *Chipumuro et al., 2014*; *Kwiatkowski et al., 2014*).

Our unbiased de novo identification of a tin-based metal cluster as a selective transcription inhibitor provides an example of a potential category of metal-based biologically active compounds that is distinct from the conventional transition metal-based POMs (*Rhule et al., 1998*). The critical role of histidine coordination demonstrated here is a direct expansion of a recent report on a nickel/ cobalt substitution in potentiating POM compounds to prevent amyloid β peptide aggregation involved in Alzheimer's diseases (*Gao et al., 2014*). With locally enriched charged residues, the TAF2 IDR binding to the metal complex may involve electrostatic interactions as well. Histidine coordination and salt bridges by themselves are unlikely to be sufficient for high-specificity interactions. However, the specific biochemical activity and the synergy with DNA (but not heparin) binding suggest a reasonable level of structural selectivity. This is consistent with our finding that a tris-Ni- nitrilotriacetic acid derivative capable of coordinating multiple histidines (*Lata et al., 2005*) can bind TAF2 in the context of human TFIID but fails to inhibit transcription under relevant concentrations (data not shown). We speculate this selectivity may be partially ascribed to the periodic nature of the rigid metal–oxygen backbone of our tin compound, which can at least discriminate secondary structures (α helices, β sheets, and random coils) of the protein target (presenting the repetitive residues with distinct periodicity). This discrimination may be sufficient to shift the equilibrium between the functional states of the dynamic IDR, as exemplified recently by a phosphorylation- induced β-barrel formation and its regulation of a disorder-to-helix transition within an IDR of the 4E-BP2 translational regulator (*Bah et al., 2015*). We thus propose that certain features of metal complexes can be harnessed in the design of new reagents targeting polar and repetitive IDRs that have largely evaded intervention by traditional organic small-molecule modulators.

## Materials and methods

### Protein purification

TFIID complexes were affinity purified from fractionated nuclear extracts with homemade monoclonal antibodies. In brief, for dTFIID, nuclear extract from 0 to ~12 hr *Drosophila melanogaster* embryos was prepared as described (*Biggin and Tjian, 1988*), fractionated by SP-sepharose resin (GE Healthcare, Pittsburgh, PA), and elution at 1 M KCl salt concentration from the SP–column was reloaded to a Q-sepharose column (GE Healthcare). A fraction eluted at 0.3 M KCl salt concentration from the Q-column was collected, immune-precipitated by a monoclonal antibody (2B2) raised against the TAF1 subunit of dTFIID, and eluted with a specific epitope peptide. The purity and integrity of the complex was verified by silver staining of sodium dodecyl sulfate-polyacrylamide gel electrophoresis (SDS-PAGE)- separated samples and protein mass spectrometry. For hTFIID, nuclear extract from HeLa cells was fractionated by P11 phosphate-cellulose resin (Whatman, Maidstone, United Kingdom) and collected at 1

M KCl salt elution, then affinity purified as described (*Liu et al., 2009*). Human TFIIH and Pol II complexes were purified from HeLa nuclear extract as described (*Revyakin et al., 2012*).

Recombinant proteins were expressed in *Escherichia coli* and purified according to manufacturer suggested methods. GST-fusion proteins were purified by Glutathione-sepharose 4B resin (GE Healthcare), and (His)$_6$-Halo fusion proteins was purified by Ni-NTA agarose resin (Qiagen, Venlo, Netherlands) to ~>90% purity in SDS-PAGE Coomassie brilliant blue-stained gels. Nuclease treatment of recombinant protein for small-molecule microarray binding was carried out on beads before elution with 0.5 U/µl DNase I (Roche, Basel, Switzerland) and 10 ng/µl RNase (Sigma–Aldrich, Milwaukee, WI) at room temperature with mixing for 1 hr in a buffer containing 50 mM Tris pH 8.0, 5 mM MgCl$_2$, 2.5 mM CaCl$_2$, 5% glycerol, and 1 mM dithiothreitol (DTT).

## Plasmid constructs and primers

All the SCP1-related constructs, except for the 'L' DNA used in *Figure 7A*, were constructed by inserting the SCP1 and mutants as described (*Juven-Gershon et al., 2006*) in place of the adenovirus E1a promoter (TATA box) of the p*G3*-BCAT plasmid (*Ryu et al., 1999*) upstream to a chloramphenicol acetyl transferase (CAT) reporter gene. The 'L' DNA template had an extra 37 base-pair insertion after the promoter. The G-less cassette DNA template containing the adenovirus major late promoter is the pML(C2AT)19Δ-50 as reported (*Szentirmay and Sawadogo, 1994*) (a gift from Dr Manabu Mizuguchi).

For the primer extension assay used to quantify transcription products, a reverse primer (5′ GCCATTGGGATATATCAACGGTGG 3′) starting at position +155 (+192 for the 'L' DNA template) relative to the transcription start site was used. For the DNase I footprinting assay, the reverse primer was used together with a forward primer (5′ CATAACCTTATGTATCATACACATACG 3′), starting at position −153, to generate a DNA template by PCR. DNA oligonucleotide primers were custom synthesized (Integrated DNA Technology, Coralville, IA). Both transcription and footprinting assays used wild-type SCP1-containing DNA unless otherwise specified.

## Small-molecule microarray

Small-molecule microarray manufacture and screening were carried out as described (*Casalena et al., 2012*). In brief, the slides were blocked in 3% (wt/vol) bovine serum albumin (BSA) (Sigma–Aldrich, Milwaukee, WI) in phosphate-buffered saline supplemented with 0.1% Tween 20 (PBST) for 1 hr at room temperature with gentle shaking, rinsed with 0.04% BSA in PBST, then incubated at 4°C for 1 hr, sequentially, with protein of interest (concentrations optimized for the best signal/noise ratio, usually 1–10 µg/ml), primary antibody (at optimized dilutions) and secondary antibody (1/1000) diluted in PBST (with or without 0.04% BSA) and washed 5 min in between with the same buffer. For the rescue of nuclease-treated protein, heparin (Sigma–Aldrich) (2 µg/ml) or a dsDNA oligonucleotide (5′ GCTTGCATGCGTACTTATATAAGGGGGTGGGGGCGCGTT 3′) (1 µg/ml) was incubated together with the recombinant protein (0.5 µg/ml). The slides were further rinsed with PBST and water briefly, spun dry, immediately scanned at 532 nm and 635 nm using a GenePix 4000B (Molecular Devices, Sunnyvale, CA) slide scanner and the images were analyzed by GenePix Pro6 software (Molecular Devices). The primary, monoclonal antibodies are either in-house raised and Protein G affinity purified from hybridoma culture supernatants (anti-dTAF4, 3E12; anti-hXPB, 30C1-1; anti-Rpb1: 8WG16) or commercially available (anti-hTAF4: BD Biosciences (Sao Paulo, Brazil) Cat #612054; Anti-GST: Sigma–Aldrich Cat #G-1160_0.5ML). Cy5-labeled secondary antibody against mouse IgG was purchased from GE Healthcare Life Sciences (Cat #PA45002). For the plots in *Figure 1C*, the strongest signals from reference dyes were removed.

Note: the retention of SnOCl$_2$ cluster on the microarray is likely mediated through coordinating interactions and/or local polymerization of the material, which will preserve similar surface features as proposed in *Figure 3—figure supplement 3E* for protein interactions.

## Surface plasmon resonance assay

The surface plasmon resonance experiments were performed using a BIACORE T100 (GE Healthcare). GST and GST tagged dTAF2(1125–1221) were captured on reference cell and active cell, respectively, through GST capturing kit (GE Healthcare BR100223) on CM5 sensor chip (GE Healthcare BR100012). For the capturing of hTAF2 (990–1199), a HaloTag Amine(O4) Ligand (Promega, Fitchburg, WI, Cat #P6741) was immobilized on all flow cells using amine-coupling chemistry on CM5 sensor chip.

A (His)$_6$-Halo-tagged reference protein (PP7 bacteriophage coat protein [*Chao et al., 2008*]) and a (His)$_6$-Halo-tagged hTAF2 (990–1199) were immobilized through Halo tag on the reference and active cell, respectively (no evidence for direct interaction between the (His)$_6$ tag and the inhibitor was observed). The inhibitor binding assay was performed in running buffer (20 mM HEPES pH 7.5, 100 mM KCl, 0.5 mM EDTA, 0.005% surfactant Polysorbate 20, and 2% DMSO) (HEPES: 4-(2-hydroxyethyl)-1-piperazineethanesulfonic acid; EDTA: ethylenediaminetetraacetic acid; DMSO: dimethyl sulfoxide) at a flow rate of 30 µl/min with 120 s of association and dissociation at 25˚C. SnOCl$_2$ or SnOCl$_2$·pyridine was prepared at concentration of 5 µg/ml in different buffers adjusted with citrate for lower pH or supplemented with imidazole at specified concentrations.

## GST pull down assay

For TAF2 IDR-DNA binding, GST fusions immobilized on glutathione sepharose 4B beads (~2 mg/ml) were briefly pretreated with 100 µg/ml RNase A in PBST in the presence of complete protease inhibitor (Roche) to remove endogenous nucleic acid (predominantly RNA) from *E. coli*. 10 µl beads were incubated in a 384-well plate with 20 ng/µl double-stranded DNA oligonucleotides suspended in 40 µl PBST for 1 hr with gentle shaking at 4˚C. 20 µl of the unbound fraction was saved and the beads were washed and re-suspended in 20 µl PBST. Both the bound and unbound were supplemented with 10 µl, 4 µg/ml ethidium bromide and scanned by Typhoon scanner (GE Healthcare) and the signals were quantified by ImageQuant TL software (GE Healthcare).

## In vitro reconstituted transcription

The procedure was as described (*Revyakin et al., 2012*) with minor modifications. The GTFs included in a standard 25 µl reaction are ~10 ng (0.1 µl) affinity purified h/dTFIID, 5 ng hTFIIB, 20 ng hTFIIE-α, 20 ng hTFIIE-β, 20 ng hTFIIF, ~5 ng TFIIH, and ~20 ng Pol II. As controls for TFIID specificity, 5 ng recombinant dTBP (his-tagged) was used. These factors were pooled and diluted in a GTF buffer (10% glycerol, 25 mM HEPES pH 7.9, 12.5 mM MgCl$_2$, 100 mM KCl, 0.1 mM EDTA, 0.01% NP40, 0.02% Tween 20, 1 mM DTT, and 100 µg/ml BSA). The chemicals or the DMSO control was dissolved with the same GTF buffer (with a final DMSO concentration of 2%), kept on ice for a few hours, spun at 15,000 rpm for 15 min, before they were used to treat the protein factors (total volume 12.5 µl) at specified concentrations at room temperature (23˚C) for 15 min. Next, 100 ng DNA in 12.5 µl DNA buffer (0.2 µl RNasin (Promega) and 8 mM spermidine in water) was added and incubated at 30˚C for 30 min for PIC assembly. NTPs were then added to 0.5 mM each to allow RNA synthesis at 30˚C for 30 min. RNA products were analyzed by primer extension and 6% urea-polyacrylamide gel electrophoresis, scanned by Typhoon scanner (GE Healthcare), and quantified by ImageQuant TL software (GE Healthcare).

For the dinucleotide synthesis monitoring abortive initiation, following modifications were made according to a previous report (*Orphanides et al., 1998*): (1) the NTP substrates only included 1 mM ATP and 1.3 µM GTP (including 0.3 µM from α-$^{32}$P labeled GTP, 3000 Ci/mmole, 10 mCi/ml, Perkin Elmer, Waltham, MA); (2) the transcription was stopped by incubation at 65˚C for 30 min, followed by 4 U shrimp alkaline phosphatase (New England Biolabs (NEB), Ipswich, MA) treatment at 37˚C for 1 hr; and (3) the end product was separated by 15% TBE-urea gel (Thermo Fisher Scientific, Waltham, MA).

For the step-wise PIC assembly test shown in *Figure 6*, GTF set 1 (6 µl) was incubated with template DNA (6 µl) for 30 min at 30˚C before the addition of inhibitor/DMSO (8 µl, in GTF buffer) to specified concentrations. After inhibitor treatment at specified concentration for 15 min at 23˚C, GTF set 2 (3 µl) was added together with 11 µl DNA buffer, and PIC assembly was finished by another incubation at 30˚C for 30 min. For the results shown in *Figure 6—figure supplement 1*, 6 µl of inhibitor/DMSO was used to get the specified concentrations, GTF set 2 was in 1 µl, and DNA or DNA buffer was 7 µl. For the comparison of inhibitor effect before or after PIC assembly (*Figure 8* and *Figure 8—figure supplement 1B*), the complete GTF mix, DNA template (100 ng DNA in DNA buffer), inhibitor/DMSO diluted in GTF buffer, and DNA buffer alone are all 6 µl each.

The colliding polymerase assay using G-less cassette DNA template has the following modifications based on a previous report (*Szentirmay and Sawadogo, 1994*): (1) the template DNA was pML(C2AT)19Δ-50 as reported (*Szentirmay and Sawadogo, 1994*); (2) ATP and UTP were 0.6 mM; CTP was 25 µM (including 0.3 µM diluted from α-$^{32}$P labeled CTP, 3000 Ci/mmole, 10 mCi/ml, Perkin Elmer); (3) no GTP was included in the reaction; (4) the reaction was stopped with 100 µl of Stop Solution (20 mM EDTA, 1% SDS, 0.2 M NaCl, 0.15 µg/µl glycogen, and 0.1 µg/µl Proteinase K), further

incubated at 55℃ for 10 min, then extracted by phenol:chloroform:isoamylalcohol (50:49:1) and precipitated with ethanol; (5) the precipitation pellet was re-suspended in 4 µl buffer (50 mM Tris pH 7.5 and 2 mM EDTA) containing 10 U/µl RNase T1 (Thermo Fisher), digested at 37℃ for 10 min, then separated by 6% polyacrylamide-urea denaturing gel.

Sarkosyl was tested in our system following previously a reported procedure (*Hawley and Roeder, 1985*). To restrict transcription to a single round, Sarkosyl was added either to a final concentration of 0.02% just before the addition of NTPs or to 0.1% immediately (within ~30 s) after the addition of NTPs.

For the two-DNA, template-commitment assay, DNA 1 (or DNA buffer only) was incubated with the complete set of protein factors in GTF buffer for 30 min at 30℃ in a total volume of 12 µl, followed by an addition of 6 µl GTF buffer (with extra TFIID as an option) and 6 µl DNA 2. After incubation at 30℃ for a specified period of time, NTPs were added and the incubation continued at 30℃ for another 30 min for RNA synthesis. Sarkosyl was added to 0.1% immediately (within ~30 s) after the addition of NTPs as an option to restrict transcription to a single, first round.

## DNase I footprinting assay

The assay is essentially as described (*Cianfrocco et al., 2013*) with modifications. In brief, a DNA fragment was generated by PCR using a primer starting at −153 (from the transcription start site) of the non-template strand, and a radioactively ($^{32}$P) labeled primer starting at the +155 of the template strand. The TFIID-promoter binding (*Figure 5C*) was carried out in 10 µl buffer containing (5% glycerol, 12.5 mM HEPES pH 7.9, 6 mM MgCl$_2$, 50 mM KCl, 50 µM EDTA, 0.05% NP40, 0.01% Tween 20, 50 µg/ml BSA, 1% DMSO, 0.5 mM DTT, ~240 ng hTFIID, and ~0.3 nM template DNA), with or without the chemical at specified concentrations, at 30℃ for 30 min. For the high-salt challenge experiments in *Figure 5—figure supplement 1*, extra KCl was added to bring to the specified final concentrations and the 30℃ incubation was extended by another 5 min. The reaction was brought to 100 µl (with 90 µl: 5% glycerol, 12.5 mM HEPES pH 7.9, 6 mM MgCl$_2$, 50 mM KCl, 50 µM EDTA, 0.5 mM DTT, and 2.5 mM CaCl$_2$) then immediately digested by pre-diluted DNase I (Worthington, Lakewood, NJ) at 30℃ for 60 s. The DNase I digestion was stopped with 100 µl of Stop Solution (20 mM EDTA, 1% SDS, 0.2 M NaCl, 0.15 µg/µl glycogen, and 0.1 µg/µl Proteinase K). The digestion products were purified, separated by 6% polyacrylamide-urea denaturing gel, and the image further processed as in the transcription assay.

The DNase I footprinting assay monitoring conformational isomerization during PIC assembly (*Figure 7A*) was carried out under conditions very similar to the transcription assay. Following are the modifications from a typical 25 µl transcription assay (1) the DNA template was 0.3 nM of the radioactively labeled PCR product (a 308-bp fragment, instead of the ~4 kb plasmid); (2) 100 ng purified yeast tRNA and 3 ng poly(dG:dC) were included in each reaction as carriers, and (3) ~40 ng TFIID was used; (4) at the end of PIC assembly, 2 µl 40 mU/µl DNase I (NEB) in a buffer (50% glycerol, 12.5 mM HEPES pH 7.9, 6.25 mM MgCl$_2$, 50 mM KCl, 0.05 mM EDTA, 0.005% NP40, 0.5 mM DTT, and 5 mM CaCl$_2$) was added, and after 30 s, the digestion was stopped and further processed as described above.

To monitor the inhibition of the conformational isomerization during PIC assembly (*Figure 7B*, lanes 1–8), GTF set 1 (30 ng hTFIID plus 5 ng hTFIIB, with or without 20 ng TFIIF plus 20 ng Pol II) in 8 µl GTF buffer was mixed with 0.6 nM radioactively labeled DNA template in 8 µl DNA buffer (0.2 µl RNasin and 8 mM spermidine) supplemented with 64 ng purified yeast tRNA and 2 ng poly(dG:dC), incubated at 30℃ for 30 min. Then, 4 µl SnOCl$_2$·pyridine solution (15 µg/ml in GTF buffer with 2% DMSO) or the control solution was added and incubated at room temperature for 15 min. Next, GTF set 2 (0.5 µl, with 20 ng TFIIF and 20 ng Pol II, or the GTF buffer only) was supplemented, together with 4.5 µl DNA buffer (supplemented with 36 ng purified yeast tRNA and 1 ng poly(dG:dC)) and the incubation was continued for another 30 min at 30℃. The final reaction mixture was subjected to digestion by 2 µl 40 mU/µl DNase I (NEB) and the end digestion product was analyzed by gel electrophoresis. The experiment in *Figure 7B* lanes 9–14 was performed slightly differently in that: (1) the TFIID amount was 10 ng and TFIIF was 2 ng; (2) the DNA buffer was replaced with (10 mM Tris pH 8.0, and 0.1 mM EDTA), and the carrier nucleic acids were eliminated; and (3) 2 µl 20 mU/µl DNase I (NEB) was used for the digestion. These two sets of conditions used in *Figure 7B*, when supplemented with other factors and reagents, both lead to optimal transcription output, and indistinguishable response to the tin(IV) oxochloride inhibitor.

## Chemical probe synthesis and characterization

The commercially supplied lead compound and its analogs were named by an abbreviated vender name followed by specific catalog number. More specifically, 'ChemDiv' stands for ChemDiv Inc. (San Diego, CA); 'Otava' stands for Otava Ltd (Vaughan, Canada); 'Princeton' stands for Princeton Biomolecular Research Inc. (Monmouth Junction, NJ); and 'LifeChem' stands for Life Chemicals Inc (Niagara-on-the-Lake, Canada). Nuclear magnetic resonance (NMR) and mass spectrometry were used by these commercial suppliers for quality control, which all indicated '>95%' purity of the intended organic compounds (note: those assays are not sensitive to inorganic metal compounds). Commercial proton-induced X-ray emission elemental analysis (*Figure 3B*) was carried out by Elemental Analysis Inc. (Lexington, KY) with 5 mg of each material and the abundance of detected elements was plotted.

Details of in-house synthesis and analysis of the original lead (1) and analog (2) compounds are available upon request. Detailed procedure for resynthesis and purification of analog compound **2** was also purchased from Princeton Biomolecular Research Inc. for the track of the tin-related activity (available upon request) and used to draw the scheme in *Figure 3—figure supplement 2A*. In-house high-resolution NMR spectroscopy was carried out on Varian 500 MHz instrument (*Figure 3—figure supplement 1C*).

In-house recreation of the TFIID inhibitory activity was initially carried out in 2 ml glass vials with 1 ml isopropanol and 5 mM $SnCl_2$, sealed with a Teflon-lined screw cap and incubated at 83°C in a heat block (*Figure 3—figure supplement 2C,D*). Samples were taken at different time points for activity analysis (dried under vacuum then re-dissolved with DMSO). Acetic acid and the original lead/analog organic molecules are not required for generating the active species. Other organic solvents like methanol and DMSO can also be used to generate the inhibitory activity.

Synthesis of the organic solvent coordinated $SnCl_4$ compounds (*Figure 3—figure supplement 3A*) was accomplished by drop wise addition of neat $SnCl_4$ to stirring liquid-coordinating compounds (DMSO, *n*-propanol, *iso*-propanol, *sec*-butanol, *iso*-butanol, acetylacetone). The obtained precipitates were vacuum filtered and quickly transferred into sealed flasks and dried under high vacuum. The rest of the simple organic and inorganic tin compounds were purchased from Sigma–Aldrich.

Oxygenation of $SnCl_2$ could be achieved by refluxing in *iso*-propanol under air atmosphere for several hours, which led to potent TFIID-specific transcription inhibitory activity (*Figure 3C*, and *Figure 3—figure supplement 3B*). In-house X-ray photoelectron spectroscopy measurement (*Figure 3—figure supplement 3C*) was completed on the Surface Science, model SSX-100 at Harvard's Center for Nanoscale Systems. The probe for the measurement was monochromatic aluminum K-alpha X-ray line with energy at 1.4866 keV. Flood gun was used throughout the entire experiment for sample surface charge compensation. The survey spectrum (0–1000 eV) scan was completed by taking the average of 4 scans with X-ray spot size at 800 μm and passing energy at 150 eV. Data were analyzed using CasaXPS program, and ratios of integrated peak areas for each individual element were used for quantification.

Alternatively, oxygenation of $SnCl_2$ could be achieved by several oxidizing agents yielding active material (*Figure 3C* and *Figure 3—figure supplement 3B*): (a) bubbling oxygen gas through a sintered glass tube into a solution of $SnCl_2$ (1 g) in tetrahydrofuran (26 ml), or water (26 ml) at room temperature for 2 hr; (b) addition of $H_2O_2$ (30% aqueous, 6 ml) to solution of $SnCl_2$ (2 g) in water (52.7 ml); (c) addition of sodium perborate ($NaBO_3$) to aqueous solution of $SnCl_2$; (d) addition of *m*-chloroperbenzoic acid (mCPBA) (58 mg) to a solution of $SnCl_2$ (64 mg) in water (1.8 ml); (e) addition of *tert*-butyl hydroperoxide (TBHP) to solution of $SnCl_2$ in dichloromethane. The oxygenation of $SnCl_2$ in tetrahydrofuran by oxygen to produce $(SnOCl_2)_X$ was reported previously (*Messin and Janierdubry, 1979*). The other oxygenation methods were our development. Over drying may compromise the biological activity, which can be restored by adding miniscule amount of water (1 μl per 1 mg of dry material) prior to DMSO dissolution.

Independently of oxygenation of $SnCl_2$, $SnOCl_2$ cluster was also prepared from tin(IV) chloride ($SnCl_4$) (*Figure 3C*) as previously reported (*Sakurada et al., 2000*): to a cooled (−20°C) 1 M solution of $SnCl_4$ in dichloromethane (10 ml, 10 mmol, Sigma–Aldrich), *bis*(trimethylsilyl)peroxide (BTSP) (1.784 g, 10 ml, Gelest) was added drop wise, and the reaction was allowed to proceed for 1 hr. The volatiles were removed in vacuum and the residue was used for the activity assay, or to complex with pyridine.

$SnOCl_2$·pyridine (*Figure 3C,D*, *Figure 3—figure supplement 3D,E*, *Figure 4C–F*, *Figures 6–8*) was prepared by dissolving the residue obtained above in ethyl acetate (10 ml), followed by adding pyridine until heavy white precipitate kept forming. The precipitate was filtered, washed with excess

ethyl acetate, and dried under high vacuum. Infrared spectrum for thus prepared compound was recorded on Bruker ALPHA FT-IR instrument and the resonances matched those published previously (*Dehnicke, 1961*; *Sakurada et al., 2000*).

## Acknowledgements

We thank Drs Dirk Trauner and Andrew Stern for comments guiding this study; Dr William Dynan for editing the manuscript; colleagues at Janelia Research Campus of HHMI and members of the Tjian Laboratory at UC Berkeley for critical reading of the manuscript; and Drs Shuang Zheng, David King, Sharleen Zhou, Arnie Falick, James Duffner, Olivia McPherson, Steve Johnston, and Zhongchun Wang for technical assistance. ZZ was a Leukemia and Lymphoma Society Fellow (2006–2009). The project was supported in part with federal funds from the National Cancer Institute's (NCI) Initiative for Chemical Genetics (ICG) under Contract No. N01-CO-12400, the NCI Cancer Target Discovery and Development (CTD$^2$) Network, under RC2 CA148399 (SLS), and R01 CA160860 (ANK).

## Additional information

### Competing interests

RT: President of the Howard Hughes Medical Institute (2009-present), one of the three founding funders of *eLife*, and a member of *eLife's* Board of Directors. The other authors declare that no competing interests exist.

### Funding

| Funder | Grant reference | Author |
| --- | --- | --- |
| Leukemia and Lymphoma Society | Postdoctoral Fellowship, 5226-07 | Zhengjian Zhang |
| Howard Hughes Medical Institute | Investigator | Robert Tjian, Stuart L Schreiber |
| Howard Hughes Medical Institute | Janelia Research Campus | Robert Tjian |
| National Cancer Institute | Initiative for Chemical Genetics, N01-CO-12400 | Stuart L Schreiber |
| National Cancer Institute | Cancer Target Discovery and Development (CTD2) Network, RC2 CA148399 | Stuart L Schreiber |
| National Cancer Institute | R01 CA160860 | Angela N Koehler |

The funders had no role in study design, data collection and interpretation, or the decision to submit the work for publication.

### Author contributions

ZZ, ZB, Conception and design, Acquisition of data, Analysis and interpretation of data, Drafting or revising the article, Contributed unpublished essential data or reagents; MMH, Conception and design, Acquisition of data, Analysis and interpretation of data, Contributed unpublished essential data or reagents; WH, H-JK, AKA, MKD, TAL, Acquisition of data, Analysis and interpretation of data, Contributed unpublished essential data or reagents; CI, Drafting or revising the article, Contributed unpublished essential data or reagents; ANK, SLS, RT, Conception and design, Analysis and interpretation of data, Drafting or revising the article

### Author ORCIDs

Zhengjian Zhang, http://orcid.org/0000-0002-2840-0837
Han-Je Kim, http://orcid.org/0000-0002-0305-259X
Stuart L Schreiber, http://orcid.org/0000-0003-1922-7558

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
