## [Decision Letter]

Thank you for submitting your work entitled “Chemical Perturbation of an Intrinsically Disordered Region of TFIID Distinguishes Two Modes of Transcription Initiation” for peer review at *eLife*. Your submission has been favorably evaluated by Jim Kadonaga (Senior Editor), a Reviewing Editor, and two peer reviewers.

The reviewers have discussed the reviews with one another and the Reviewing Editor has drafted this decision to help you prepare a revised submission.

The reviewers find the work interesting; however, they have problems with the initiation/reinitiation assay. Furthermore, Reviewer #2 has additional points that we believe will have to be addressed in a revised version.

Reviewer #1:

The authors report the discovery of a TFIID inhibitor that blocks the formation of PICs. Evidence is presented for the interaction of the compound with a low complexity region of TAF2 that unexpectedly stabilizes binding of TFIID to promoter DNA. Using a defined set of initiation factors and Pol II, with and without Sarkosyl, the authors conclude that the compound does not inhibit reinitiation. The data are of high quality and demonstrate that the metal complex does inhibit transcription in vitro, likely through TFIID, however the claim that reinitiation is not affected will require further proof. The manuscript is well written, but could be improved by eliminating unnecessary data.

The dual template experiment (Figure 7) demonstrates that the template(s) is in excess over at least some of the factors, TFIID is limiting, and that Sarkosyl inhibits transcription after it is added (7A). The tin compound inhibits transcription to the same relative extent in assays with or without Sarkosyl addition (7B). My main concern is the assumption that the difference between minus and plus Sarkosyl is due to reinitiation. The original Hawley paper used it to block subsequent initiation in a crude system and the mechanism was assumed to be disruption of protein-protein and/or protein-DNA interactions which Sarkosyl does. The fully formed PIC is fairly resistant to low concentrations, but subsequent PIC formation is blocked. The amount of Sarkosyl is critical since it will disrupt preformed PICs. This is especially important in a defined system that does not have high protein concentrations like the extracts did in the original Hawley paper. Why was it added after NTPs? Initiation takes only seconds. How long after NTPs were added was the Sarkosyl added? Why were the reactions carried out for 30 minutes? To make statements about reinitiation (repeated use of a single promoter), an alternative method is needed to demonstrate that reinitiation is actually occurring under the conditions and factor concentrations used. One way to do this is to use a G-less cassette with continuous labeling and direct analysis of transcripts as was done in Szertirmay and Sawadogo (1994). Reinitiation is indicated in such an experiment by seeing transcripts of different lengths corresponding to Pol II that has stacked up behind the first polymerase initiated. Without such evidence I am not convinced that reinitiation is occurring and this impacts the important interpretation of Figure 7. Also the conclusion that reinitiation is not affected is hard to reconcile with the idea that the compound stabilizes TFIID binding in an inactive conformation. After initiation the compound would be expected to bind and inhibit the next round. If reinitiation is occurring in the absence of Sarkosyl what factor is responsible for resistance to the inhibitor?

Overall, this is an interesting study (except for the description of the identification of the inhibitor) that has significant merit. A better demonstration that reinitiation is not inhibited by the compound would improve the significance of the conclusions that could be drawn. If reinitiation is really not affected, determining the identity of the resistance factor would be exciting.

Reviewer #2:

In this paper the authors report the discovery of a class of novel inhibitors of RNA polymerase II. These tin-based compounds apparently interact with an intrinsically disordered region of the TAF2 subunit of TFIID. If the inhibitors act before TFIID is stably associated with the template, transcription is very significantly reduced. The authors show that the footprint of TFIID on a TATA-containing promoter is altered near the INR in the presence of the inhibitor. They also show in a Sarkosyl-based reinitiation assay that the effects of the inhibitor appear to be confined to the initial round of transcription, with subsequent rounds unaffected. The authors conclude that this inhibitor affects the initial productive engagement of the polymerase with the template, but not the case of polymerase entering additional rounds of transcription.

I am impressed with the amount of clever work that the authors performed to track down the actual inhibitor. There is clear potential here for novel mechanistic insights into PIC assembly, initiation and very early elongation through the use of these tin-based compounds. However, I have significant reservations about the paper as it now stands.

First, it is critical to establish at what stage of transcription the inhibitor actually acts. Transcription in this paper was assayed by primer extension, usually with a primer around +150. Is polymerase able to begin transcription but is then blocked at some earlier step point? Given where the differences are in the footprints, it is tempting to imagine a blockade at initiation or clearance. For example, the initial downstream expansion of the transcription bubble might be blocked. Fortunately this would be easy to test – does the inhibitor have any effect in an abortive initiation assay? If the initial few bonds can be made in the presence of the inhibitor, how long can the transcripts become? If no RNA is made at all, can the transcription bubble form on inhibitor-treated complexes when ATP is added?

The authors interpret the footprint alterations resulting from the inhibitor as evidence for an enhanced or stabilized binding of IID to the promoter. I don't think this interpretation is justified – TFIID binding as judged by overall protection in the footprint doesn't change significantly with the inhibitor. What has changed is the alignment of (presumably) TAF2 with the INR, which in turn changed the template accessibility to DNase over this region. The localized nature of these changes provokes questions specifically about transcription bubble formation, initiation and clearance, as I just noted. It would have been useful to see if these changes persisted even when IIB and IIF were present, since these factors partially rescue transcription from the effect of the inhibitor. Another point about the footprints – as best as I can tell the TATA box is not protected. Can the authors comment on that?

Finally, a central point of the paper is the differential effect of the inhibitor on re-initiation versus the first round of transcription. I am afraid I am skeptical of the assay. The key assumption is that Sarkosyl at 0.1% prevents the assembly of new PICs and thus confines transcription to a single round. It is not widely appreciated, but Pol II PICs do not all initiate immediately once NTPs are added. I know from my own experience that, for example, transcription for 5 min with cold NTPs still allows robust new synthesis of labeled RNAs by Pol II PICs when labeled precursors are spiked in during the next 5 min. Critically, in these experiments the RNAs were very short (<20 nt), so by definition the observed transcription in the second labeling period could not have come from promoters which were already transcribed once in the initial 5 min. I suspect that the extra transcription seen when Sarkosyl is left out simply represents the failure of the otherwise late-starting complexes to initiate. This isn't simply based on my own experience – the Yudkovsky et al. paper cited here shows that TFIID-based complexes isomerize slowly to transcriptional competence, with a half time of 10-15 min. Does Sarkosyl affect that isomerization?

The novel inhibitor identified here could provide important new insights into Pol II initiation mechanisms, but in my view more should be done to characterize the inhibitor's effects.

[Editors' note: further revisions were requested prior to acceptance, as described below.]

Thank you for resubmitting your work entitled “Chemical Perturbation of an Intrinsically Disordered Region of TFIID Distinguishes Two Modes of Transcription Initiation” for further consideration at *eLife*. Your revised article has been favorably evaluated by Jim Kadonaga (Senior Editor), a Reviewing Editor, and two peer reviewers. The manuscript has been improved but there are some remaining issues that need to be addressed before acceptance, as outlined below.

In light of the following comment by Reviewer #2 we would request one small revision: “My biggest concern focuses on how the proposed mechanism is described. The authors assume (Figure 9, for example) that Pol II is recruited to the inhibited complex but somehow cannot function. However, they don't actually demonstrate that Pol II is recruited – couldn't the inhibitor work to change TFIID's conformation so that Pol II cannot join the nascent PIC? I think an argument in this direction can be made based on the right-most panel of Figure 7. It would seem that the downstream (+16 through about +40) footprint changes characteristic of the complete complex still appear when Pol II is added after the inhibitor, but the TATA to +1 changes are reduced. This seems, at least to me, to suggest that Pol II is present in the inhibited complex even though it cannot access the transcription start site.”

The Reviewing Editor agrees with this reviewer and would like the authors to also entertain this possibility when they describe the possible model of action of the inhibitor.

---

## [Author Response]

Reviewer #1: *My main concern is the assumption that the difference between minus and plus Sarkosyl is due to reinitiation. The original Hawley paper used it to block subsequent initiation in a crude system and the mechanism was assumed to be disruption of protein/protein and/or protein/DNA interactions which Sarkosyl does. The fully formed PIC is fairly resistant to low concentrations, but subsequent PIC formation is blocked*.

As suggested by the reviewer, we have now performed continuous labeling experiments using the G-less cassette DNA template, and observed transcripts of different lengths that provides independent and complementary evidence for transcription reinitiation in our system. We also confirmed that reinitiation is resistant to the inhibitor (Figure 8). We have now included additional controls for this experiment and the Sarkosyl experiments in Figure 8—figure supplement 1.

*The amount of Sarkosyl is critical since it will disrupt preformed PICs. This is especially important in a defined system that does not have high protein concentrations like the extracts did in the original Hawley paper*.

We have performed titrations of Sarkosyl at different stages of transcription initiation as tested in the original Hawley and Roeder paper (1985, JBC, Figure 2) and observed very similar dose responses with our highly purified system (Figure 8—figure supplement 1). Basically 0.02% Sarkosyl added at the beginning is sufficient to inhibit both initiation and reinitiation; a pre-assembled preinitiation complex (PIC) may be resistant to 0.02% Sarkosyl, but completely inhibited by 0.04% Sarkosyl (this is about 2 fold more sensitive than their 1987 paper, which might reflect some differences in our system and/or experimental procedures). On the other hand, after the addition of ribonucleoside triphosphate (NTP) substrates that allow productive elongation, the system becomes resistant to a broad range of Sarkosyl, up to 0.64% as tested.

*Why was it added after NTPs? Initiation takes only seconds*.

As mentioned above, if added right before NTPs, the concentration range for Sarkosyl to only prevent reinitiation is very narrow (in fact 0.02% may not be sufficient to prevent reinitiation, while 0.04% would inhibit the function of pre-assembled PIC), which are conditions that we felt were not robust and could compromise the reproducibility of our experiments. On the other hand, when added immediately after NTPs, a broad range of Sarkosyl concentrations from 0.04% to at least 0.64%, give the same result of preventing reinitiation. Therefore we chose to add 0.1% Sarkosyl within 30 seconds after NTPs addition to restrict transcription to a single round.

How long after NTPs were added was the Sarkosyl added?

The answer is within ∼30 seconds. This much incubation time is sufficient for the formation of the first 1∼2 phosphodiester bonds which will render much stronger Sarkosyl resistance (Hawley and Roeder, 1985, JBC, Figure 6).

Why were the reactions carried out for 30 minutes?

This allows elongation (and reinitiation in the absence of Sarkosyl). Under our experimental setup, this 30 min incubation time might allow some additional de novo PIC assembly (which can explain the 1.5∼1.7 fold difference between lanes 1 and 2, or between lanes 5 and 6 in Figure 8—figure supplement 1), but the majority of transcription in this period is from reinitiation (comparing lanes 1-4 with lanes 5-8 of the same panel). Reducing this time may prevent additional de novo PIC assembly, at the cost of some reinitiation (as is shown in the Szentirmay and Sawadogo, 1994, NAR paper Figures 1 and 2).

*If reinitiation is occurring in the absence of Sarkosyl what factor is responsible for resistance to the inhibitor?** *

As illustrated in the new Figure 7, the chemical appears to block an isomerization of the TFIID-promoter complex that is required for full engagement of Pol II. The factor that makes reinitiation resistant to the chemical inhibitor is the TFIID complex that becomes isomerized during the first round of transcription, which allows reinitiation to by-pass this inhibitor-sensitive stage. It is also possible that some other factor(s) may facilitate maintenance of this isomerized functional state of TFIID that we have not yet identified.

Reviewer #2:

*First, it is critical to establish at what stage of transcription the inhibitor actually acts*.* *

We have examined the synthesis of the first dinucleotide and found that transcription is inhibited at or before this step (Figure 6). We further demonstrated that functional PIC assembly is blocked at the stage of Pol II engagement by footprinting assays (new Figure 7). Since we detected the action of the inhibitor at these very early stages of transcription initiation well before the involvement of TFIIE and TFIIH (i.e. promoter melting), we have not pursued the potential for the inhibitor to also influence promoter melting and elongation.

*The authors interpret the footprint alterations resulting from the inhibitor as evidence for an enhanced or stabilized binding of IID to the promoter. I don't think this interpretation is justified*.

This concern is well appreciated, so we have further investigated the mechanisms of inhibition using additional footprinting assays to monitor potential conformational changes during PIC assembly. In new footprinting data included in the revised manuscript, we have detected strong global (not “localized”) changes in the protection patterns consistent with conformational changes and “isomerization” over an extended region of the promoter that is arrested by the inhibitor (Figure 7). Please also see our response to the same concern from Reviewer #1 on Figure 5.

*It would have been useful to see if these changes persisted even when IIB and IIF were present, since these factors partially rescue transcription from the effect of the inhibitor*.

Actually, TFIIB doesn’t rescue transcription from the inhibition, while TFIIF does so only partially (Figure 6). Since TFIIF usually functions together with Pol II, the engagement of which completely rescued transcription from inhibition and correlated tightly with the dramatic “global” isomerization at the footprinting assay, we haven’t focused on the “local” changes in these new experiments. Instead, we have performed the step-wised perturbation assays described in Figure 7, adding the inhibitor after TFIID and TFIIB, but before TFIIF and Pol II. The results we obtained were consistent with arrest of the proposed conformational isomerization required for Pol II engagement during assembly of a functional PIC (before promoter melting).

*Another point about the footprints – as best as I can tell the TATA box is not protected. Can the authors comment on that?** *

It has been previously reported that TFIID fails to efficiently protect the TATA box at the super core promoter (Juven-Gershon et al., 2006, Nat. Meth., 3:917), and that TFIIA can facilitate a structural transition to cause TATA box protection (Cianfrocco et al., 2013, Cell, 152:120). In our transcription assays, TFIIA had no detectable effect. However, we noticed that TATA box protection became increasingly evident upon the addition of TFIIB and TFIIF, and complete protection occurred once Pol II was included in the system, which is consistent with PIC assembly in the absence of TFIIA (Figure 7).

*Finally, a central point of the paper is the differential effect of the inhibitor on re-initiation versus the first round of transcription. I am afraid I am skeptical of the assay*.

To address this reservation, we have now performed an independent assay to verify the occurrence of reinitiation in our system and its resistance to the inhibitor (Figure 8).

*I suspect that the extra transcription seen when Sarkosyl is left out simply represents the failure of the otherwise late-starting complexes to initiate*.

In addition to the independent validation of reinitiation, we have also performed titrations of Sarkosyl at different time points to compare with the original Hawley and Roeder (1985 JBC) results. We observed very similar responses to Sarkosyl as previously reported (Figure 8—figure supplement 1). In that JBC paper (Figure 6), the formation of the first 1∼2 phosphodiester bond(s) is sufficient for high Sarkosyl resistance, which is mostly completed within half a minute (no difference from 20-minute incubation). Therefore we believe the slow-starting complexes observed by Reviewer #2 could either reflect variations among experimental systems, or correlates with a stage after the synthesis of the first 2∼3 nucleotides (but before ∼20 nucleotides).

*Does Sarkosyl affect that isomerization?** *

The TFIID-DNA complex isomerization described by Yakovchuk et al. (the Yudkovsky paper is on reinitiation) (also see the new Figure 7) occurs during PIC assembly (which should be near completion with our 30-minute incubation), before the addition of NTPs, and thus should not be affected by Sarkosyl treatment.

[Editors' note: further revisions were requested prior to acceptance, as described below.]

*My biggest concern focuses on how the proposed mechanism is described. The authors assume (*Figure 9*, for example) that Pol II is recruited to the inhibited complex but somehow cannot function. However, they don't actually demonstrate that Pol II is recruited- couldn't the inhibitor work to change TFIID's conformation so that Pol II cannot join the nascent PIC? I think an argument in this direction can be made based on the right-most panel of*
Figure 7
*– it would seem that the downstream (+16 through about +40) footprint changes characteristic of the complete complex still appear when Pol II is added after the inhibitor, but the TATA to +1 changes are reduced. This seems, at least to me, to suggest that Pol II is present in the inhibited complex even though it cannot access the transcription start site*.

We agree with Reviewer #2 and thank him/her for the cautious and accurate assessment of the potential Pol II interaction with downstream promoter DNA in the presence of the inhibitor. We have modified the corresponding Results section (see the subsection “The inhibitor arrests an isomerization step required for full Pol II engagement”) and Figure 9 legend (panel B) to incorporate this helpful suggestion.